# Transcriptional Regulatory Network of GA Floral Induction Pathway in LA Hybrid Lily

**DOI:** 10.3390/ijms20112694

**Published:** 2019-05-31

**Authors:** Wenqi Li, Yubing Yong, Yue Zhang, Yingmin Lyu

**Affiliations:** Beijing Key Laboratory of Ornamental Germplasm Innovation and Molecular Breeding, China National Engineering Research Center for Floriculture, College of Landscape Architecture, Beijing Forestry University, Beijing 100083, China; lwq15001311318@163.com (W.L.); yongyubing@163.com (Y.Y.); zhangyue2017@bjfu.edu.cn (Y.Z.)

**Keywords:** GA floral induction pathway, LA hybrid lily, transcriptome RNA-Seq, GCN, qRT-PCR

## Abstract

Background: The LA hybrid lily ‘Aladdin’ has both excellent traits of Longiflorum hybrids and Asiatic hybrids—such as big and vivid flower, strong stem, high self-propagation coefficient, and shorter low temperature time required to release bulb dormancy in contrast to Oriental hybrids. A genome-wide transcriptional analysis using transcriptome RNA-Seq was performed in order to explore whether there is a gibberellin floral induction pathway in the LA hybrid lily. Subsequently, gene co-expression network analysis was used to analyze the possible interactions of key candidate genes screened from transcriptome data. At the same time, a series of physiological, biochemical, and cultivation tests were carried out. Results: The content of five endogenous hormones changed sharply in the shoot apex during the treatment of 200 mg/L exogenous gibberellin and the ratio of ABA/GA_3_ dropped and stayed at a lower level after 4 hours’ treatment from the higher levels initially, reaching a dynamic balance. In addition, the metabolism of carbohydrates in the bulbs increase during exogenous gibberellin treatment. A total of 124,041 unigenes were obtained by RNA-seq. With the transcriptome analysis, 48,927 unigenes and 48,725 unigenes respectively aligned to the NR database and the Uniprot database. 114,138 unigenes, 25,369 unigenes, and 19,704 unigenes respectively aligned to the COG, GO, and KEGG databases. 2148 differentially expression genes (DEGs) were selected with the indicators RPKM ≥ 0, FDR ≤ 0.05 and |log2(ratio)| ≥ 2. The number of the upregulated unigenes was significantly more than the number of the downregulated unigenes. Some MADS-box genes related to flowering transformation—such as *AGL20*, *SOC1*, and *CO*—were found to be upregulated. A large number of gibberellin biosynthesis related genes such as *GA2ox*, *GA3ox*, *GA20*ox, *Cytochrome P450*, *CYP81*, and gibberellin signal transduction genes such as *DELLA*, *GASA*, and *GID1* were significantly differentially expressed. The plant hormones related genes such as *NCED3* and sugar metabolism related genes such as *α-amylase*, *sucrose synthase hexokinase*, and so on were also found expressing differentially. In addition, stress resistance related genes such as *LEA1*, *LEA2*, *LEA4*, *serine/threonine protein kinase*, *LRR receptor-like serine/threonine protein kinase*, *P34 kinase*, *histidine kinase 3* and epigenetic related genes in DNA methylation, histone methylation, acetylation, ubiquitination of ribose were also found. Particularly, a large number of transcription factors responsive to the exogenous gibberellin signal including *WRKY40, WRKY33, WRKY27, WRKY21, WRKY7, MYB, AP2/EREBP, bHLH, NAC1, NAC2*, and *NAC11* were found to be specially expressing. 30 gene sequences were selected from a large number of differentially expressed candidate genes for qRT-PCR expression verification (0, 2, 4, 8, and 16 h) and compared with the transcriptome expression levels. Conclusions: 200mg/L exogenous GA_3_ can successfully break the bulb’s dormancy of the LA hybrid lily and significantly accelerated the flowering process, indicating that gibberellin floral induction pathway is present in the LA lily ‘Aladdin’. With the GCNs analysis, two second messenger G protein-coupled receptor related genes that respond to gibberellin signals in the cell were discovered. The downstream transport proteins such as *AMT*, *calcium transport ATPase*, and *plasma membrane ATPase* were also discovered participating in GA signal transduction. Transcription factors including *WRKY7*, *NAC2*, *NAC11*, and *CBF* specially regulated phosphorylation and glycosylation during the ubiquitination degradation process of *DELLA* proteins. These transcription factors also activated in abscisic acid metabolism. A large number of transcription factors such as *WRKY21*, *WRKY22*, *NAC1*, *AP2*, *EREB1*, *P450*, and *CYP81* that both regulate gibberellin signaling and low-temperature signals have also been found. Finally, the molecular mechanism of GA floral induction pathway in the LA hybrid lily ‘Aladdin’ was constructed.

## 1. Introduction

The genus *Lilium* belongs to the family Liliaceae and comprises over 115 species. Lilies are one of the most popular perennial bulbous plants throughout the world and they occupy a prominent position in horticulture as a cut flower, potted plant, and garden plant at present. However, dormancy in lily bulbs is the most binding constraint to their commercial production. The effects and scope of low temperature change greatly depend on species or varieties [1]. Proper chilling treatments (0–8 °C) are required for at least 6–8 weeks to break the bulb’s dormancy for the majority of Oriental hybrid lilies [2]. Some Asiatic hybrids only need 4 weeks to release bulbs’ dormancy [3]. For some Asiatic hybrids such as ‘Cordelia’, 12 °C is a low enough temperature to break dormancy [4]. In contrast, low O_2_ concentration without any low temperature could promote releasing the dormancy of *Lilium longiflorum* ‘Hinomoto’ [5]. The LA hybrids were derived from back crossing with the F1 hybrid of Longiflorum x Asiatic hybrid to an Asiatic hybrid (Genome composition of LA hybrids is indeed ALA) in most cases. The LA hybrid Lily ‘Aladdin’ has many desirable characteristics, such as large vivid flowers, elegant fragrance, and strong stems like other LA hybrids. Most importantly, this variety also has strong self-reproducibility of multiplying a lot of sub-bulbs and a lower requirement of accumulation of cold for breaking dormancy [6]. In view of the above advantages, The LA hybrid lily is a very useful garden cultivar. We speculate that other floral induction pathways may exist and play important roles in the floral transition in LA hybrid lily except the vernalization pathway. A lot of evidence has verified that a certain concentration of gibberellins can break the dormancy of lily bulbs to speed up the flowering process on a cultivation and physiological level. A study found that the germination time of Lanzhou lily bulbs treated with GA_3_ reduced from 105 days to 90 days [7]. A study revealed that gibberellin treatment of 24 h broke dormancy in *Lilium speciosum* ‘Star Gazer’ and ‘Snow Queen’ [8]. A previous research indicated that a concentration of gibberellic acid (1 mg/L) reduced the level of dormancy in *Lilium speciosum* [9]. Another study verified a similar result more in depth—that 1 mg/L gibberellic acid for six weeks, or cold treatment for six to eight weeks, had significant positive effects on the dormancy breaking of in vitro bulblets of Lilium cultivars [10]. Germination significantly improved in Wild Lily with 100 mg/L of GA_3_ treatment (*Lilium sargentiae* Wilson) [11]. When the bulbs are treated with 200 mg/L gibberellin during the treatment of 56 days and 5 °C chilling and protected culture, flowering of 50% of the plants is 92 days and 10 days earlier than conventional culture and chilling combined protected culture of the control, the flower rate is 97.3% [12].

GA dormancy-breaking and floral induction pathway have since been demonstrated to be an important way to release dormancy and modulate flowering in many species and varieties of lilies as mentioned above. The biosynthesis and signal transduction of GAs are two major effective phases in GA pathway, which have been described previously in Arabidopsis and many other higher plants. In recently years, a number of factors involved in the GA pathway have been identified and characterized. In brief, GAs biosynthesis from *GGPP* can be divided into three stages. (1) formation of ent-kaurene under the catalysisi of *CPS* and *KS*; (2) formation of *GA12* via two multifunctional cytochrome P450 monooxygenases, *KO* and *KAO*; and (3) production of bioactive GAs under the catalysis of *GA20ox* and *GA3ox*. Meanwhile, the bioactive GAs can be deactivated by *GA2ox. CPS, KS, KO*, and *KAO* enzymes are usually encoded by single or few genes. Unlikely, *GA20ox, GA3ox*, and *GA2ox*—which play catalysis roles in the third stage—are encoded by multiple differentially expressed genes. Likewise, GA biosynthesis pathway genes are under feed-back control. It includes inhibition of some *GA20ox* and *GA3ox* genes expression and activation of some GA2ox genes expression [13].

GA signaling transduction pathway in Arabidopsis is initiated through its binding to the *GA INSENSITIVE DWARF1* (*GID1*) receptor. This allows subsequent interaction between *GID1* and *DELLA* proteins (*GA INSENSITIVE*[*GAI*], *REPRESSOR OR GAI3*[*RGA*], *RGA-LIKE1*[*RGL1*], *RGL2*, *RGL3*), which are transcriptional repressors when unbound by *GID1*. In the presence of gibberellins, the stable *GID1-GA-DELLA* complexes are recognized by the F-box protein *SLEEPY1* (*SLY1*) based *SCFSLY1* complex, which ubiquitylates the DELLA proteins and causes their degradation by the *26S proteasome* [14]. DELLAs are members of the plant-specific *GRAS* protein family and a key factor of GA signaling, acting as a repressor of GA signaling. *DELLA RGL2* appeared to switch between positive and negative regulation of *GID1* expression in response to dormancy-breaking treatments [15]. However, research on the downstream components of *DELLA* is limited at present. Several mechanisms about *DELLA*-mediated transcriptional control have been demonstrated. DELLAs could act as transcriptional co-activators of *phytochrome interacting factors* (*PIFs*), belonging to the *basic helix–loop–helix* (*bHLH*) transcription factor family [16]. *DELLAs* regulate seed germination by interaction with *PIF1* [17]. Another recent study revealed that *DELLA* proteins interact with *FLC* to repress floral transition. The C-terminus of MADS domain of *FLC* and the LHRI domains of DELLAs are required for the interaction [18]. In the vernalization pathway, *FLC* is a core transcriptional repressor that downregulates the expression of *SOC1* and *FT*, both genes have been proven to be major floral regulators and they further activate a series of genes involved in the formation of floral meristems. The interaction between *FLC* and *DELLA*s may integrate various signaling inputs in flowering time control. Another study indicated that GA-induced expression of *FT* is dependent on *CO* to modulate flowering under LDs [19]. It has been demonstrated that most *GASA* genes are transcriptionally regulated by GAs. The relationship between *GASA* and *DELLA* was also investigated. A previous study revealed that *AtGASA6* function downstream of *RGL2* [20]. Other additional studies also demonstrated that GA regulates the expression of *GASA1, GASA4, GASA6*, and *GASA9* through *DELLA* proteins [21]. Moreover, a previous study showed that *GASA4* and *GASA6* are in general upregulated by growth hormones (auxin, BR, cytokinin, and GA) and downregulated by stress hormones (ABA, JA, and SA), indicating a role of *GASA4* and *GASA6* in hormone crosstalk [22]. *MiRNA159*, which was negatively regulated by *DELLA*, represses *MYB DOMAIN PROTEIN33* (*MYB33*) and thus controls the expression of *LFY* [23]. More down-stream signal components of *DELLA* protein are required to be separated.

The transition from dormancy to germination is a vital dynamic morphological, physiological, and biochemical process, which controls the flowering time. For developing alternative and more quickly floral induction techniques compared to vernalization, the effects of lily bulb pre-treatments with gibberellic acid (GA_3_) on dormancy-release were investigated in our study. However, despite their importance, few studies have been carried out to date on the transcription factors of GA pathway in lily. A study indicated that lots of transcripts related to *DELLA, GA3ox*, and *GA2ox* in the GA pathway of the Oriental lily differentially expressed in the process of hypothermia [24]. Therefore, in an effort to advance our understanding of the response to exogenous GA_3_ application in lily and the mechanism of the GA floral induction pathway, we carried out RNA-seq transcriptome and RT-PCR to investigate gene expression patterns in the meristems of lily bulbs after treatments with GA_3_ for different times and identify GA-related genes on dormancy breaking and floral induction. Recently, Illumina sequencing technology, RNA-sequencing (RNA-Seq) technology has become the most powerful and popular tool for plants that lack reference genome information, which is less costly, more efficient, and more accurate and sensitive profiles than other techniques. However, the majority of genes encoded in the genome remain uncharacterized. One promising approach to improve our understanding of how these genes may function in lily is through gene co-expression analysis (GCA). Genome-wide gene expression data from DNA microarrays in plants has proved useful for defining correlated expression patterns between genes using pairwise similarity metrics such as Pearson’s correlation coefficient, r, and subsequent genome-scale reconstruction of gene co-expression networks (GCN). Genes are usually represented as ‘nodes’, whilst the lines linking individual nodes, or ‘edges’, represent pairwise relationships between nodes. A collection of densely connected nodes represents a ‘cluster’ and the entire collection of nodes, edges and clusters forms the co-expression ‘network’. Often, co-expressed genes within a cluster are expected to be functionally related to genes with a similar expression pattern. This ‘guilt-by-association’ approach has become a powerful tool for transcriptional regulatory inference and understanding the evolution of transcript expression within and between plants. Markov cluster algorithm (MCL) has been widely used to partition the complex gene co-expression network of plants in to defined functional clusters. Using genome-wide guide and graph clustering of GCNs, systematic assessments of clusters were performed using a combination of GO enrichment analysis, gene expression information, and literature searches.

## 2. Results

### 2.1. Endogenous Phytohormone Changes and Carbohydrates Status

Germination depends on regulation of phytohormones, including gibberellic acid (GA), abscisic acid (ABA), ethylene, and auxin. Among them, ABA and GA are proved to be key regulators, which play antagonistic roles in seed germination [25]. From the ratio of the other four plant hormones to the GA_3_ content, it can be seen that the ratio of ABA/GA_3_, IAA/GA_3_ decreased rapidly, and reached an equilibrium after 4 h. The ratio of ZR/GA_3_ decreased slowly and reached a balance at 4 h. The ratio of JA/GA_3_ has remained in a steady and slow decline (Figure 1). The results indicated that the endogenous hormone content will change rapidly within 4 h by the exogenous GA_3_ treatment, and after 4 h, it will reach a certain relative equilibrium state. From the results, we found the endogenous phytohormone—GA_3_, ABA, IAA, and JA change intensely after the 200mg/L exogenous GA_3_ treatment except ZR. The content of GA_3_ was 26.34ng/g. FW in the 16 h is 6 times that of 0 h. The content of endogenous GA_3_ continued to rise during the exogenous GA_3_ treatment. The content of ABA was 77.62ng/g. FW of 0 h is 94.34ng/g. FW at 4 h, 120.24ng/g. FW in the 16 h is almost 2 times that of the 0 h value. However, in the 2 h, the content of ABA was slightly reduced. The decrease of content at 2 h indicated that there was antagonistic relationship between ABA and GA_3_, but as the content of GA_3_ increased, the content of ABA increased, indicating a synergistic relationship between the two. The content of JA was 37.32ng/g. FW at 4 h, which is 3 times of the original 13.31ng/g. FW in the 0 h. However, the content of JA had dropped in the 8 h and 16 h measures. The content of IAA was 95.85ng/g. FW in the 0 h was 49.37ng/g. FW in the 2 h, 64.41ng/g. FW in the 4 h and 77.82ng/g. FW in the 16 h. The content of ZR was 10.21ng/g. FW in the 0 h, 5.35ng/g. FW in the 2 h, 7.07ng/g. FW in the 4 h was 8.36ng/g. FW in the 16 h (Figure 2).

The results showed that the content of soluble sugar and starch in the inner scales did not change much after treatment with exogenous GA_3_. The soluble sugar and starch content decreased at 2 h, and gradually increased after 4 h, followed by soluble sugar content. It showed a slow downward trend. The starch content decreased again at 8 h, and it rose back to the original level at 16 h. This indicates that the soluble sugar in the inner scales may be continuously consumed, and the continuous supply is from the inner layer starch conversion, but most should come from the supplement of soluble sugar in the outer scales. It can be seen from the changes of soluble sugar and starch in the outer scales that the soluble sugar content decreased rapidly at 2 h, which may be transferred to the inner scales, resulting in the recovery of soluble sugar content in the inner scales within 4 h. The starch content is significantly reduced and continuously converted into soluble sugar. The soluble sugar content in the outer scales was basically the same as the starch content. In the outer scales, the sugar content drops rapidly and the starch content subsequently decreased. The soluble sugar content in roots was significantly higher than the content in starch, and the starch content was basically unchanged during the treatment. The soluble sugar content increased slightly at 2 h, indicating that the soluble sugar in the outer scales is also transferred to the roots. After 2 h, the soluble sugar content in the outer scales decreased sharply, and the soluble sugar content in the 4 h roots also decreased slowly. In the case of the soluble sugar in the outer scales continued to decrease, the soluble sugar content in the 8 h roots also decreased significantly, and in the 16 h the outer scales also showed a decrease. The starch converted into soluble sugar, which causes the soluble sugar content in the root to rise to a large extent. It also indicates that the transfer direction of soluble sugar is from the outer scales turn to the inner scales (Figure 3).

### 2.2. Illumina Sequencing Data, De Novo Assembly, and Gene Annotation

To enrich the number of genes involved in our transcriptome, cDNA samples were extracted from total RNA isolated from SAM for three libraries (0, 2, and 4 h samples treated by GA). Approximately 47 million raw reads for the 0 h sample and 104 million for the latter two samples (2 h and 4 h samples) were obtained. After data cleaning and stringent quality checks, 139,175,054 raw reads containing a total of 22.8 Gb nucleotides were obtained. The average read size, GC percentage, and Q20 percentage were 519 bp, 43.87%, and 96.60%, respectively. 153,850 contigs were assembled with an average length of 573 bp based on the high-quality reads. All de novo assembly yielded 124,041 unigenes with an average length of 599.77 bp (Table 1 and Table 2). 48,927 (39.44% of the total) and 48,725 (39.28% of the total) had significant similarity to known proteins in the NR and Uni-Prot database, respectively (Table 3). To demonstrate the quality of sequencing data, 30 unigenes were selected and 30 pairs of primers were designed for qRT-PCR, and then the products were confirmed by biological Sanger sequencing.

### 2.3. Functional Classification

To further evaluate the effectiveness of our annotation process and the completeness of our transcriptome library, the annotated sequences were randomly searched for genes with COG classifications. Of 48,715 hits, 98,366 sequences had a COG classification (Table 4). Among the 25 COG categories, the cluster for ‘signal transduction mechanisms’ (17,076, 17.36%) represented the largest group, followed by ‘cytoskeleton’ (15,655, 15.92%) and ‘general function prediction only’ (8685, 8.8%). The following categories, ‘defense mechanisms’ (393, 0.40%), ‘extracellular structures’ (393, 0.40%), and ‘cell motility’ (23, 0.02%), represented the smallest groups (Figure 4).

The GO assignments were also used to classify the functions of the predicted lily genes. Based on sequence homology, 124,041 sequences can be categorized into 56 groups (Figure 5A). In each of the three main GO classifications, the ‘metabolic process (28.18% in BP), ‘cell part’ (22.81% in CC), and ‘binding’ (48.13% in MF) terms were dominant, respectively. We also found a high percentage from the ‘cellular process’ (23.36% in BP), ‘organelle’ (16.63% in CC), and ‘catalytic activity’ (41.1%% in MF) categories, but few from ‘biological adhesion’ (0.02% in BP), ’behavior’(0% in BP), ‘virion part’(0.05% in CC), ’extracellular matrix component’ (0% in CC), ‘metallochaperone activity’ (0.01% in MF), and ‘protein tag’ (0% in MF) (Figure 5B). The GO analysis indicated that the identified genes were associated with various biological processes.

Based on a comparison against the KEGG database using BLASTx with an Evalue cutoff of <10^−5^, of the 124,041 unigenes, 19,704 (15.89%) had significant matches in the database and were assigned to 325 KEGG pathways. The most representative pathways were ‘metabolic pathways’ (4337 members), ‘biosynthesis of secondary metabolites’ (2285 members), ‘biosynthesis of antibiotics’ (1147 members), ‘microbial metabolism in diverse environments’ (995 members), and ‘ribosome’ (504 members) (Appendix A). These annotations provided an efficient resource for investigating pathways, functions, and processes involved in lily dormancy-breaking. The plant hormone signal transduction pathway (ko04075) was also included in the KEGG categories.

### 2.4. Identification and Expression Analysis of DEGs

A RNA-Seq experiment was conducted using samples applied by GA_3_ for 0, 2, and 4 h of LA hybrid lily and mapped the resulting reads to our reference transcriptome to determine the DEGs. The following DEGs were filtered with an RPKM≥ 0, FDR ≤ 0.05 and |log_2_(ratio)| ≥2. The 2 h and 0 h libraries were compared and 7782 genes were found, a total of 3965 upregulated and 3817 downregulated genes were detected between the two libraries. The number of the upregulated genes and the downregulated genes are almost the same. There were 5475 upregulated and 3955 downregulated genes between the 4 h and 0 h libraries, 3657 upregulated and 2157 downregulated genes between the 4 h and 2 h libraries (Figure 6). This suggests that the number of DEGs between 4 h and 0 h is larger than that between the 2 h and 0 h libraries, while the difference between the 4 h and 2 h libraries is the smallest of the three. We also found that the numbers of upregulated genes were higher than the numbers of downregulated genes after GA_3_ treatment in LA lily.

Filtering with an RPKM≥ 2, FDR ≤ 0.05 and |log2(ratio)| ≥ 2, 2148 genes were differentially expressed among the three stages based on the transcriptome profiling results. A total of 1002 upregulated and 173 downregulated genes were detected between the 2 h and 0 h libraries. The upregulated genes in this group accounted for 46.6% of the total DEGs, almost the half of the total DEGs. The downregulated genes in this group accounted for 8% of the total DEGs. The total number DEGs between the 4 h and 0 h libraries accounted for 70.5% of the total DEGs, this is greater than the group of the 2 h and 0 h libraries which accounted for 54.6% of the total DEGs. However, the number of DEGs between the 2 h and 0 h libraries is more than half of the total DEGs which suggests that genes expressed rapidly in 2 h in response to GA_3_ treatment. A total of 1252 upregulated and 264 downregulated genes were detected between the 4 h and 0 h libraries, accounting for 58.3% and 12.3% of the total DEGs, respectively. A total of 381 upregulated and 53 downregulated genes were detected between the 4 h and 2 h libraries, accounting for 17.7% and 2.5% of the total DEGs, respectively. Moreover, the number of the upregulated genes and the downregulated genes has a huge gap in each comparison group. As well as the above analysis of DEGs filtered with an RPKM ≥ 0, FDR ≤ 0.05 and |log2(ratio)| ≥ 2, the number of DEGs (RPKM ≥ 2, FDR ≤ 0.05 and |log2(ratio)| ≥ 2) between 4 h and 0 h libraries is the largest, followed by the 2 h and 0 h comparison group, the number of DEGs in 4 h and 2 h comparison group is the smallest of all. That means, in LA hybrid lily, transcript abundance changed dramatically from the stage without gibberellins treatment to the gibberellins treatment stages in which the GA response genes could be induced and expressed strongly. Meanwhile, the differentially expressions of these key genes determine the rapid response to the exogenous GA_3_ in an instant in the lily (Figure 6).

Through the Venn diagram analysis of the 2148 DEGs, three genes(contig_19485, Contig3551, contig_43047) were identified in three pairwise transcriptome comparisons, one of the genes is *LEA4* (*Late embryogenesis abundant group 4*, contig_43047), the other two genes are uncharacterized, all three genes were constantly and dramatically up/downregulated between 0 h and 4 h, suggesting these genes may always play an important role in the whole process of the GA3 treatment. Furthermore, 414, 586, and 174 DEGs were found only in 2 h vs. 0 h, 4 h vs. 0 h, and 4 h vs. 2 h transcriptomes, accounting for 19.3%, 27.3%, and 8.1% of the total DEGs, respectively. 714 DEGs, accounting for 33.2%, were found both in 2 h vs. 0 h and 4 h vs. 0 h transcriptomes, in which 106 DEGs were characterized, such as *LEA (Late embryogenesis abundant protein*, contig_123089), *NAC1 (NAC protein* 1, Contig12404, Contig5556), *MYB TF* 2 (*Myb transcription factor 2*, Contig 5472), *WRKY TF 40, 33, 27, 21, 7* (*Transcription factor WRKY 40, 33, 27, 21, 7*, Contig16070, Contig10474, contig_112644, contig_100827, contig_112645, contig_36867, Contig6963, Contig10395, contig_91975, contig_58912), *AP2/EREBP TF* (*AP2/EREBP transcription factor superfamily protein*, contig_15479), gene encoding the *xyloglucan endotransglucosylase/hydrolase protein* (Contig 11584), and *P34 kinase* (Contig 2523). 213 DEGs, accounting for 9.9%, were found both in 4 h vs. 0 h and 4 h vs. 2 h transcriptomes, in which 18 DEGs were characterized, one of which is the gene encoding the *diacylglycerol kinase protein* (Contig 4850, contig_40020). 44 DEGs, accounting for 2%, were found both in 2 h vs. 0 h and 4 h vs. 2 h transcriptomes, in which only two genes were characterized, one of which is the gene encoding the *PEPC* (*Phosphoenolpyruvate carboxylase*, contig_98336) (Figure 7).

With an algorithm developed from expression trend profiles of DEGs (RPKM ≥ 2, FDR ≤ 0.05 and |log_2_(ratio)| ≥2), the 2148 DEGs were classified into eight expression-trend profiles. The results showed that the profile 6, of which the genes’ expression increased sharply from 0 h to 2 h and then kept stable from 2 h to 4 h accounted for the largest proportion, containing 1041 genes. It was closely followed by the profile 4, including 271genes, of which the genes’ expression remained temporarily almost the same from 0 h to 2 h and subsequently rose from 2 h to 4 h. The third is profile 7, which presented that the genes’ expression rose constantly from 0 h to 4 h, containing 242 genes. The next was the profile 2, including 195 genes, which showed the expression pattern of genes decreasing initially from 0 h to 2 h and then increasing from 2 h to 4 h. The following was the profile 0 which showed continuous downtrend in expression, including 150 genes. The sixth was the profile 1, including 107 genes, which showed a downward trend from 0 h to 2 h and kept a stable trend from 2 h to 4 h. The seventh was the profile 3, including 74 genes. The last was the profile 5, which contained 68 genes. Their expression trends were shown in Figure 8 below. The heatmap of all the DEGs showed the consistent expression trends. On one side, some genes were downregulated from the 0 h to 2 h stages, but upregulated obviously from 2 h to 4 h stages. On the other side, it was showed that some genes increased expression at the 2 h stage, but decreased transcript abundance at the 4 h stage. Some genes were immediately expressed at the initial stage of 0 h, while others were upregulated subsequently (Figure 9).

### 2.5. Functional Enrichment Analysis of DEGs

The GO assignments were used to classify the functions of the 2148DEGs (Filtered with an RPKM ≥ 2, FDR ≤ 0.05 and |log2(ratio)| ≥ 2) after 4 h GA_3_ treatment. In each of the three main GO classifications, the ‘metabolic process (29.74% in BP), ‘cell and cell part’ (23.23% and 23.23% in CC), and ‘binding’ (45.19% in MF) terms were dominant, respectively. We also found a high percentage from the ‘cellular process’ (19.93% in BP), ‘organelle’ (15.65% in CC), and ‘catalytic activity’ (41.9% in MF) categories. The single-organism process (17.11% in BP), biological regulation (7.94% in BP), response to stimulus (6.71% in BP) cannot be ignored. The GO analysis indicated that the identified genes were associated with various biological processes (Figure 10). The functional GO enrichment analysis of 2148 DEGs was shown in the figure below (Appendix A).

Based on a comparison against the KEGG database using BLASTx with an Evalue cutoff of <10^−5^, the 2148 DEGs had significant matches in the database and were assigned to 83 KEGG pathways (Appendix A). The functional KEGG enrichment analysis of 2148 DEGs was shown in the figure above (Appendix A). The most representative pathways were ‘amino sugar and nucleotide sugar metabolism’ (16DEGs), ‘glycolysis/gluconeogenesis’ (16 DEGs), ‘phenylpropanoid biosynthesis’ (13 DEGs), ‘starch and sucrose metabolism’ (13 DEGs), ‘biosynthesis of amino acids’ (13 DEGs), ‘carbon metabolism’ (13 DEGs), ‘alpha-linolenic acid metabolism’ (12 DEGs), ‘flavonoid biosynthesis’ (11 DEGs), ‘glutathione metabolism’ (11 DEGs), ‘phenylalanine metabolism’ (10 DEGs), ‘glycerophospholipid metabolism’ (10 DEGs), ‘plant-pathogen interaction’ (9 DEGs), ‘Zeatin biosynthesis’ (8 DEGs), ‘circadian rhythm-plant’ (8 DEGs), ‘fatty acid degradation’ (8 DEGs), ‘flavone and flavonol biosynthesis’ (7 DEGs), ‘ubiquinone and other terpenoid-quinone biosynthesis’ (7 DEGs), and ‘plant hormone signal transduction’ (7 DEGs). These annotations provided an efficient resource for investigating pathways, functions, and processes involved in lily dormancy-breaking (Figure 11).

### 2.6. Differential Expression Analysis of Key Genes

Transcription factors: A large number of transcription factors were found, including *WRKY 85* (Contig 5897), *WRKY 40* (Contig 16070), *WRKY 33* (contig_100827, Contig10474, Contig6963, contig_36867), *WRKY 27* (contig_58912), *WRKY 21* (contig_112644, contig_112645), *WRKY 7* (contig_91975, Contig13143), *WRKY* transcription factor (Contig10395, contig_74229), heat shock transcription factor *Heat shock A2* (contig_147190, contig_68192), *Heat Shock A-2b* (Contig10086), transcription factor *AP2/EREBP* (contig_15479, contig_64211, contig_6641), *AP2-like* (contig_31023), *AP2-domain containing transcription factor* (Contig16130), transcription factor *NAC1* (Contig12) 404, Contig5556), *NAC11* (contig_47533), *NAC-domain containing transcription factor* (contig_47532, Contig2543, contig_122890, contig_5277), transcription factor *MYB* (Contig10978, contig_107841, Contig2955), *R2R3 MYB* (Contig806contig_135561, Contig10033, contig_112030), *MYB2* (Contig5427), *MYB5* (Contig9242), *MYB98* (contig_75829), MADS-box transcription factor *AGL20* (contig_14431), transcription factor *CBF* (Contig9007), transcription factor *EREB1* (Contig9291), transcription factor *zinc referring to Zinc finger CCCH domain-containing protein 32* (contig_19526), ethylene response transcription factor *Ethylene responsive transcription factor 2b* (contig_65834), and so on (Appendix A).

Floral transition related genes: flowering integrator *AGL20/SOC1* (contig_14431, contig_16411, contig_70281, Contig1815, contig_44220, Contig12365, Contig1888, contig_10322, contig_10670), contributing flower factor *CO* (Contig5082), transcription factor *bHLH* (contig_14876, contig_55931, contig_66658, contig_80665, contig_91633), and transcription factor *bHLH25* (contig_95131) that may regulate the CO gene (Appendix A).

Hormone-related genes: *cytochrome P450* (contig_495, Contig11622), *cytochrome P450 CYP81* (contig_57244, Contig81), *CYP8* (Contig1222), *cytochrome P450 89A2* (Contig15092), *DELLA* protein *(DWARF8)* (Contig3159), *GASA*-like protein (contig_1561, contig_1562), *cold-regulated LTCOR12* (contig_67925), cold-regulated gibberellin regulatory protein *1LTCOR12* (contig_133755), *gibberellin 20 oxidase* (contig_58931), *gibberellin 3-oxidase* (Contig7586, Contig15483), *gibberellin 3-oxidase 2* (contig_30501), *gibberellin 2-oxidase 4* (contig_14262), *gibberellin 2-β-double oxygenase GA2ox* (contig_79537), *9-cis epoxy carotenoid dioxygenase NCED3* (contig_127138, contig_82115), *tryptophan biosynthesis 1* (Contig 3534), *cytokinin oxidase/dehydrogenase 6 isoform 1* (Contig 14170), *SAUR-like auxin-responsive family protein* (contig_105313, contig_96011, contig_96020), *auxin response GH3 family protein* (contig_120322), *1-aminocyclopropane-1-carboxylic acid oxidase* (Contig7564), E*3-ubiquitin-protein ligase UPL6* (Contig13942), *E3 ubiquitin E3 ubiquitin-protein ligase BRE1-like protein* (Contig13942), *ring finger protein* (contig_84742), *heat shock protein* (Contig1525), *heat shock protein 70* (contig_8855), *cytoplasmic class I small heat shock protein 3B* (contig_2205 4), *chaperone protein* (contig_87062, Contig15475), *calmodulin-binding protein* (Contig15102), *calmodulin-like protein* (contig_127735), *60S ribosomal protein L31* (contig_121296), 4*-coumaric acid-CoA ligase 9* (contig_37094), p*hotosystem I P700 chlorophyll a apolipoprotein A1* (Second_Contig 1938), and so on (Appendix A).

Carbohydrate metabolism-related genes: *glucose 6-phosphate dehydrogenase* (Contig6729), *sucrose synthase* (contig_22392, Contig4527), *fructose diphosphate aldolase* (contig_6925), *Xylose isomerase* (contig_21412), *endoglucanase* (Second_Contig1564), *xyloglucan endo-glucokinase/hydrolase* (Contig15472, Contig11584, contig_62659), *glycosyltransferase* (contig_2821, contig_33603, contig_61457, contig_83432, contig_93519, contig_98102, Contig12157, Contig12635, Contig13119, Contig13294, Contig15238, Contig9400, Contig9565, Second_Contig294), *β-hexosaminidase* (contig_46355), *hexyl glycosyltransferase* (contig_27832, contig_8764, Contig5280), *anthocyanin 5,3-o-glucosyltransferase* (Contig 13404), *soluble acid invertase* (contig_68521), and so on (Appendix A).

Stress-resistant related genes: *glutathione transferase* (contig_44762, Contig8504, contig_101889, contig_53305, contig_53306, contig_58173, contig_79316, Contig11293, Contig13715), *thioredoxin* (contig_3971, contig_3972, contig_98142), *actin depolymerization factor* (contig_43718), *disease-resistant family protein LRR family protein* (Contig602, contig_3570), *late embryogenesis-rich protein 1 (LEA1)* (Contig15947, contig_24694), *late embryogenesis-rich protein 2* (contig_99782), *late embryogenesis-rich protein 4* (contig_43047), *serine/threonine protein kinase* (Contig5432, Contig17002, Contig13127, Contig16418, Contig16448, contig_30231, contig_75279, contig_99357), *LRR receptor-like serine/threonine protein kinase* (Contig16507), *receptor-like protein kinase* (contig_56469), *P34 kinase* (Contig2523), *histidine kinase 3* (Contig1188), and *mitogen-activated protein kinase* (contig_53060) (Appendix A). 

Histone modification related genes: methylation-related gene *methyltransferase* (Contig13028), *adenosine homocysteine* (contig_61885, contig_101207), *Orcinol O-Methyltransferase methyltransferase* (contig_37751, contig_53117), *ubiquitination-modification-related genomic protein H2A* (contig_1031, contig_2045), *histone H3* (contig_19960, contig_425), ADP-ribosylation-related gene *poly[ADP-ribose] polymerase gene* (contig_59490, Contig16608), *NADPH enzyme gene* (contig_33047), and so on (Appendix A).

### 2.7. Identification and Expression of MCL Modules in GCNs

We used the MCL to detect biologically related modules in GCN, which resulted in 25 modules containing 1226 DEGs (the total number of DEGs in GCN is 1296) with cluster size ranging from 15 to 202 nodes. The simple parameters of constructed GCN and HRR values between each co-expressed genes are presented in table below, respectively (Table 5). Through gene expression analysis of DEGs in each detected module, we found that most modules were mainly composed of upregulated genes, while Module 4, Module 10, Module 14, Module 18, and Module 23 were mainly composed of downregulated genes (Figure 12).

### 2.8. Functional Enrichment Analysis, Key Genes Mining, and Expression of MCL Modules

GO annotation is one of important steps in the GCN analysis to understand the biological functions of each detected module [26]. First, we carried out a GO classification of the GCN (Figure 13). Then, we carried out a GO enrichment analysis using AgriGO analysis tool, which revealed that modules were mainly involved in metabolic processes (GO:0008152), cellular process (GO:0009987), single organism process (GO:0044699), biological regulation (GO:0065007), response to stimulus (GO:0050896), and response to gibberellin (GO:0009739). The identification of over-represented and/or statistically significant biological processes can further reveal the functional features of each detected module. In this study, almost all modules were enriched to metabolic process, response to stimulus, biological regulation, response to gibberellin, and six of them (Module 1, 2, 3, 5, 12, and 14) were statistically significant, indicating that Module 1, Module 2, Module 3, Module 5, Module 12, and Module 14 are the GAs response related function modules.

In gene co-expression analysis, functional consistency among the highly inter connected genes present in the same modules at a given cut-off is very important for gene mining. In this study, functional enrichment analysis has resulted in significant enrichment of 6 modules (Module 1, 2, 3, 5, 12, and 14) containing genes principally involved in various kinds of stimulus response, biological regulation, and metabolic processes.

Module 1 consisted of 201 gene nodes and is highly enriched with regulation of transcription (GO:0006355, FDR = 0.00137), biosynthetic process (GO:0044283, FDR = 0.01), peptidyl-tyrosine modification (GO:0018212, FDR = 0.028), catabolic process (GO:0019320, FDR = 0.029), abscisic acid metabolic process (GO:0009687, FDR = 0.045). 89% of genes in the module are upregulated by GA_3_, while rest of the genes is downregulated (Figure 12). A total of 26 genes associated with various kinds of biological process (FDR ≤ 0.05), half of which were annotated genes in this module. The genes are labelled in different colors and co-express the network to show the interaction between genes, as shown in the figure below (Figure 14). The expression of genes in Module 1 was presented through the heatmap (Appendix A).

Red represents genes enriched to transcriptional regulation (contig_104756, contig_136393, contig_58865, contig_75642, contig_84469, contig_88830, Congtig11200 genes mentioned above are not annotated in the transcriptome contig_47532 *NAC2*, contig_47533 *NAC11*, contig_68192 *Heat shock transcription factor A2*, contig_91975 transcription factor *WRKY7*, Congtig10395 transcription factor *WRKY*, Congtig9007 transcription factor *CBF*).

Yellow represents enrichment to organisms Genes in anabolic processes (contig_112053, contig_14091, Congtig5132, Congtig9730 genes mentioned above were not annotated in transcriptome analysis, contig_12731 *Myristylglycerol phosphate dehydratase*, contig_49990 *3-phosphonic acid 1-carboxylate Vinyl transferase*, contig_58629 *acyl-CoA oxidase*, Congtig1084 *phenylalanine ammonia lyase*).

Blue represents a gene enriched for phosphorylation (contig_30231 *serine/threonine protein kinase*, Congtig16507 *LRR receptor-like serine/threonine protein kinase*).

Pink represents a gene involved in glucose metabolism (Congtig6131 *2,3-independent bisphosphoglycerate*). Purple represents the related gene enriched in the process of abscisic acid metabolism (contig_46737 *phosphoglycerate mutase PGAM glycase*).

Module 2 consisted of 185 gene nodes and is highly enriched with metabolic process (GO:0008152, FDR = 0.0014), oxidation-reduction process (GO:0055114, FDR = 0.045), cation-transport (GO:0006812, FDR = 0.04), G-protein coupled receptor signaling pathway (GO:0007186, FDR = 0.001), gibberellic acid mediated signaling pathway (GO:0009740, FDR = 0.007), histone phosphorylation (GO:0016572, FDR = 0.03). 83% of genes in the module are upregulated by GA_3_, while rest of the genes are downregulated (Figure 15). A total of 66 genes associated with various kinds of biological process (FDR ≤ 0.05), 23 of which were annotated genes in this module (Appendix A). The expression of genes in Module 2 was presented through the heatmap (Appendix A).

Among them, red represents the related genes enriched in the metabolic process (contig_12281, contig_14965, contig_15328, contig_16874, contig_35259, contig_35269, contig_35863, contig_42673, contig_44982, contig_50831, contig_59742, contig_60833, contig_65348, contig_68281, Congtig13057, Congtig1569, Congtig4504, Congtig5569, Congtig6338, Congtig6362, Congtig7663, Congtig7910, Congtig8189, Congtig9369, Congtig1598, Congtig1706 transcriptome analysis in these genes were not being functional annotation, contig_11208 *S -acyltransferase*, contig_33219 *4- coumarate-CoA ligase -A-*, contig_37306 *regulation of cyclin-dependent kinases*, contig_37751 *moss melanin methyltransferase*, contig_3974 *translator correlators*, contig_41795 *serine/threonine protein phosphatase*, contig_63014 *bipolar kinesin KRP130*, Congtig12157 *glycosyltransferase*, Congtig14307 *chitinase A*, Congtig15472 *xyloglucan*, Congtig4527 *sucrose synthase*, Congtig6250 *4- coumarate-CoA ligase -A-*, Congtig6482 *patatin*).

Yellow represents the related genes enriched in the redox process (contig_21048, contig_25874, contig_34849, contig_9424, Congtig2210, Congtig7357, Congtig8594, Congtig9510, Congtig9532 These genes are not functionally annotated in the transcriptome, contig_1057 *polyphenol*, contig_33047 *NADPH adrenaline redox Enzyme*, Congtig6729 *Glucose-6-phosphate dehydrogenase*).

Blue represents the relevant genes enriched into the cation transport process (contig_48589, Congtig9824 has no functional annotation in the transcriptome, contig_33848 *Ammonium transporter*, contig_58025 *plasma membrane ATPase*, Congtig12384 *calcium transport ATPase*).

Green represents the related gene enriched in the G protein pathway (contig_40020, Congtig4850 *diacylglycerol kinase*).

Pink represents enrichment Related genes in gibberellin signal transduction (not functionally annotated in the Congtig 8847 transcriptome).

Orange representing genes enriched in histone phosphorylation (contig_46610 transcriptome Commented by the function).

Module 3 consisted of 113 gene nodes and is highly enriched with regulation of cellular process (GO:0050794, FDR = 0.02), oxidation-reduction process (GO:0055114, FDR = 2.12E−05), dephosphorylation (GO:0016311, FDR = 0.025), response to oxygen-containing compound (GO:1901700, FDR = 0.034) and response to cold (GO:0009409, FDR = 0.039). 80% of genes in the module are upregulated by GA_3_, while rest of the genes is down regulated (Figure 12). A total of 46 genes associated with various kinds of biological process (FDR ≤ 0.05), 21 of which were annotated genes in this module. Label genes with different colors and co-expressing networks to show inter-genetic interactions (Figure 16). The expression of genes in Module 3 was presented through the heatmap (Appendix A).

Among them, red represents genes enriched in biological process regulation (contig_15387, contig_588269, contig_79589, Congtig10445 are genes without functional annotation in transcriptome analysis, contig_112644 transcription factor *WRKY21*, contig_112645 transcription factor *WRKY22*, contig_6641 transcription factor *AP2-EREBP*, Congtig12404 *NAC1*, Congtig14497 *non-specific serine/threonine protein kinase*, Congtig9291 *EREB1*).

Yellow represents genes enriched in the redox process (contig_45556, contig_79536, contig_79556, Congtig10276, Congtig10705, Congtig16363, Congtig9314 are all genes with no functional annotation in transcriptome analysis, contig_12385 *alcohol dehydrogenase 1A*, contig_495 *Cytochrome P450*, contig _68686 *Peroxidase*, Congtig10117 *1-Aminocyclopropane-1-carboxylic acid oxidase*, Congtig11622 *cytochrome P450*, Congtig487 *alcohol dehydrogenase*, Congtig75641-*aminocyclopropane-1-carboxylic acid oxidase*, Congtig81 *cytochrome P450 CYP81*).

Green represents the gene enriched in dephosphorylation (contig_80931, Congtig13868 transcriptome has no gene function annotation, contig_41048 *cytoplasmic 5’-nucleotidase III*).

Pink represents genes enriched in response to oxygenates (Congtig12279, Congtig3551 transcriptome analysis has no relevant functional annotations).

Blue represents genes enriched in response to low temperatures (no functional annotations in theCongtig15068 transcriptome, Congtig11584 *xyloglucan endotransglucosyl hydrolase*).

Module 5 consisted of 66 gene nodes and is highly enriched with cellular protein modification process (GO:0006464, FDR = 0.04), response to hormone (GO:0009725, FDR = 0.03), response to stress (GO:0006950, FDR = 0.04), polysaccharide metabolic process (GO:0005976, FDR = 0.008). 82% of genes in the module are upregulated by GA_3_, while rest of the genes is downregulated (Figure 12). A total of 30 genes associated with various kinds of biological process (FDR ≤ 0.05), 14 of which were annotated genes in this module. Label genes with different colors and co-expressing networks to show inter-genetic interactions (Figure 17). The expression of genes in Module 5 was presented through the heatmap (Appendix A).

Among them, red represents genes enriched in cellular protein modification (Congtig16549, Congtig16937, Congtig16547, Congtig12167, Congtig5432, contig_76328, contig_77935, contig_25080 is a gene with no functional annotation in transcriptome analysis, Congtig16418 *serine/threonine protein kinase*).

Yellow represents a gene enriched in response hormone (contig_106340 transcriptome analysis functionally annotated genes, Congtig12166 *pec*tin gene, Congtig16130 transcription factor *AP2*).

Pink represents genes enriched in response stress (Congtig6326 *pectin*, contig_50764 *hexosyltransferase*).

Blue represents Genes enriched in polysaccharide metabolism (Congtig15847, Congtig5541 transcriptome analysis, no functionally annotated genes, Congtig16162 *peroxidase*).

Module 12 consisted of 34 gene nodes and is highly enriched with regulation of biological processes (GO: 0050789), vegetative growth to reproductive growth (GO: 0010228), L-phenylalanine metabolism (GO: 0006558). 50% of genes in the module were upregulated (Figure 12). Labeled genes with different colors and co-expressing networks to show inter-genetic interactions (Figure 18). The expression of genes in Module 12 was presented through the heatmap (Appendix A).

Red represents genes enriched for biological process regulation (first no functionally annotated genes in the Congtig10652 transcriptome analysis, contig_55686, contig_90494 transcriptome analysis without functional annotation Gene, contig_58912 transcription factor *WRKY27*, contig_98142 *thioredoxin* gene). 

Yellow represents the gene enriched into vegetative growth to reproductive growth (contig_66570, contig_55519, Congtig10396 are all transcriptome analysis without functional annotation gene). 

Blue represents enrichment to L-phenylalanine metabolism (contig_55519 *phenylalanine ammonia lyase*).

Module 14 consisted of 31 gene nodes and is highly enriched with protein phosphorylation (GO: 0006468), cell recognition (GO: 0008037), response to gibberellin (GO: 0009739). All genes in the module were downregulated (Figure 12). Label genes with different colors and co-expressing networks to show inter-genetic interactions (Figure 19). The expression of genes in Module 14 was presented through the heatmap (Appendix A).

Red represents genes enriched for protein phosphorylation (Congtig14322, Congtig12128, Congtig17002 are genes with no functional annotation in transcriptome analysis, contig_9602 is a *serine/threonine protein kinase*). Yellow represents rich genes that are integrated into cell recognition (Congtig12121, Congtig4110 are genes that have no functional annotation in transcriptome analysis). Blue represents a gene that is enriched in response to gibberellin (a gene that has no functional annotation in transcriptome analysis atCongtig9196).

### 2.9. qRT-PCR Verifing Gene Expression Profiles

To verify the gene expression in our Illumina sequencing analyses, 20 DEGs containing two candidate genes (*Gibberellic acid insensitive, GAI* contig 1509; *GA-insensitive dwarf mutant1a, GID1a* contig_18004) related to the gibberellin signal transduction were selected for qRT-PCR using samples of 0, 2, and 4 h stages originally used for RNA-Seq, all of which are known to be related to Gibberellins signal, including the genes encoding *GA2ox* (*gibberellin 2-oxidase*, Contig_79537), *DELLA* (*Dwarf8*, Contig3159), *CBF* (*C-repeat binding factor*, Contig9007), *LTCOR12* (*gibberellic acid-stimulated in Arabidopsis, GASA*, Contig_133755), *ER* (*estrogen receptor*, Contig_43645), *AP2-EREBP* (*Apetala2/ethylene responsive element binding protein*, Contig_6641), *SAUR-like* (*small auxin-up RNA-like*, Contig_96020), *auxin-responsive GH3* (*auxin-responsive Gretchen Hagen 3*, Contig_120322), *DELLA* (*GAI*, Contig1509), *GID1a* (Contig18004) and so on. The Ct values of the *LoTIP* (tonoplast intrinsic protein) in all samples ranged from 24.0 to 28.0. All transcripts showed the same expression pattern as the in silico differential analysis results from high-throughput sequencing.

These genes were selected for their key roles in regulating GA signal transcription, endogenous plant hormones responses, and cold responses. The results presented in Figure 20 showed that the expression levels of *DELLA* (*Dwarf8* Contig3159, *GAI* Contig1509) decreased sharply in 2 h and 4 h samples than in the 0 h sample, which indicated that the two genes may play key roles in GA signal response and may delay the floral transition. The expression of transcription factor genes *CBF* (Contig9007) and *ER* (contig_43645) increased sharply in the 2 h sample than in the untreated sample (0 h), which indicated that these TF genes may play important roles in GA signal response and transduction process. The expression of *GA2ox* (contig_79537) was higher in 2 h and 4 h stages than the 0 h stage, which indicated that *GA2ox* has a passivation effect to gibberellin. *AP2-EREBP* (contig_6641), *SAUR-like* (contig_96020), and *auxin-responsive GH3* (Gretchen Hagen 3,contig_120322) are related to endogenous hormone. The expression of *AP2-EREBP* (contig_6641) continued to drop sharply in 2, 4, and 8 h, and increased a little in 16 h, which was still less than half of the expression in 0 h. The expression of *SAUR-like* (contig_96020) upregulated dramatically in 2 h, and continued to downregulate in 4 h and 8 h, following a sharp recovery in 16 h, which was almost 11 times the original expression level in 0 h. The *auxin-responsive GH3* (contig_120322) continued to upregulate dramatically except a sharp fall in 8 h. The cold-regulated gibberellin-regulated gene *LTCOR12* (*GASA* contig_133755) expressed more in 2 h and 4 h stages than 0 h and expressed highest after being treated by. GA for 16 h, which showed an obvious delay in expression contrast to the genes mentioned above. The expression of the gene *GAI* (Contig1509) also encoding *DELLA* protein and a gibberellin signal receptor gene *GID1a* (contig_18004) did not show differential in the first three stages (0, 2, and 4 h), but decreased significantly in the 16 h stage. We predicted that the bulb has played an important role in for the LA hybrid lily adaptation to GA and releasing the dormancy.

Another 10 genes were selected for qRT-PCR (Figure 21) in GCN modules. Three genes in Module 2:Contig8847 is a key gene that responds to gibberellin by gene co-expression clustering and GO function enrichment, but it is not annotated in transcriptome analysis. The expression level increased slightly in 2 h, and the content in 4 h increased sharply. In the qRT-PCR, the content decreased continuously during the treatment time. *Calcium transport ATPase-related gene* (Contig12384) showed a significant increase in transcriptome results. It also showed an upward trend in qRT-PCR. The transcription factor *AGL20* (*Agamousclike 20*, contig_14431) was significantly upregulated in the transcriptome results, and the expression of 4 h increased sharply. In the qRT-PCR, the expression increased sharply at 2 h, decreased slightly at 4 h, and increased sharply at 8 h. Rising expression trends.

Two genes in Module 5: contig_106340 is also a key gene in response to gibberellin analyzed by gene co-expression clustering and GO function enrichment, but it is not in transcriptome analysis. In the transcriptome data, the expression level increased sharply at 2 h, but decreased at 4 h, but it still showed upregulated expression. In the qRT-PCR, it increased sharply at 2 h, decreased at 4 h, and increased sharply at 8 h, and the whole expression was upregulated. The transcription factor *AP2* (Contig16130) was upregulated in the transcriptome results and was upregulated in 2h, and decreased in 4 h, and upregulated in the whole. In the qRT-PCR results, the expression was upregulated at 2 h and 4 h, and downregulated at 8 h.

Two genes in Module 1: the transcription factor *WRKY7* (contig_91975) has a sustained downregulated expression in the transcriptome, and there is a slight increase in qRT-PCR results in 2 h, followed by downregulation. The transcription factor *NAC11 (NAM-ATAF-CUC 11*, contig_47533) continued to be upregulated in the transcriptome data, it was upregulated in the qRT-PCR results until 4 h, and downregulated in 8 h.

There are three genes in Module 3: the transcription factor *WRKY21* (contig_112644) is continuously upregulated in the transcriptome data and is downregulated in the qRT-PCR analysis at 4 h and upregulated in 8 h. The transcription factor *NAC1* (Contig12404) continued to be significantly upregulated in transcriptome data and was downregulated in qRT-PCR analysis. The transcription factor *EREB1* (*ineversible electrical break down 1*, Contig9291) continued to be significantly upregulated in the transcriptome results, with a slight downregulation in qRT-PCR results.

## 3. Discussion

### 3.1. Carbohydrate Metabolism During the Exogenous GA3 Treatment

Studies have found that exogenous sucrose treatment can increase the convulsion rate of lily and make flower buds appear early [27]. Adding proper amount of sucrose can induce tissue culture plants to flower in advance [28]. The transport and accumulation of sucrose in apical meristem may be an important signal for flower induction [29,30]. Sucrose can cause the flowering branches of the peach blossom to bloom in advance and prolong the flowering period [31]. Significant differential expression of genes involved in in starch and sucrose metabolism pathway (ko00500) by the KEGG enrichment analysis were found in the transcriptome data of this assay, including sucrose synthase, fructose diphosphate aldolase, xylose isomerase, anthocyanin 5,3-o glucose. The expression of related genes such as basal transferase, hexosyltransferase, and glycosyltransferase was significantly upregulated in the study (Figure 22).

The content of soluble total sugar and starch in LA lily bulbs also changed significantly. The soluble sugar content of the inner scales decreased rapidly and then rose rapidly. The starch content of the inner scales also in the same trend, but the overall content did not decrease significantly. The soluble sugar and starch content of the outer scales continued to decrease, indicating the soluble sugar of the outer scales is continuously transferred to the inner scales. The starch in the outer scales was continuously converted into soluble sugar and directed to the inner scales. After exogenous GA_3_ treatment, the carbohydrates in the bulbs were significantly active in metastasis and metabolism, and it was also a physiological response to the treatment of exogenous GA_3_ to relieve the dormancy of the bulbs.

### 3.2. Diverse Plant Hormones Interaction to Release the Dormancy

Germination depends on regulation of phytohormones, including gibberellic acid (GA), abscisic acid (ABA), ethylene, and auxin. Among them, ABA and GA_3_ are proven to be key regulators [25]. The interaction between GA_3_ and ABA is complex, both synergistic and antagonistic. Studies have shown that cereal seed embryos release gibberellin into aleurone cells during germination, induce a large amount of *α-amylase* transcription, hydrolyze starch and protein, and supply energy and nutrients for germination [32]. Studies have shown that ABA inhibits transcription of the alpha-amylase gene [33]. *PKABA1* is a Ser/Thr protein kinase induced by ABA, which regulates the expression of ABA inhibiting *α-amylase* gene; *GAMYB* is a gibberellin-inducible *MYB* transcription factor that activates the α-amylase gene [34]. Barley studies have found that *SLR1*, abscisic acid, and *PKABA1* inhibit the transcription of *GAMYB* and α-amylase genes [35]. Studies in Arabidopsis showed that both GA_3_ and ABA induced the accumulation of miR159, and the target gene of mi R159 was *MYB33* mRNA [36]. Therefore, it is speculated that these two hormones have both antagonistic and synergistic effects. It has also been found that there is a *WRKY* transcription factor induced by ABA in rice to regulate the gibberellin response. These transcription factors inhibit GA-induced *alpha-amylase* gene transcription [37], but they are related to *PKABA1*. The relationship is still unclear. The above is the explanation of the interaction between GA_3_ and ABA in the molecular layer. In the study, we also found some genes involved in plant hormone signal transduction pathway (ko04075) by the KEGG enrichment analysis, indicating that ABA, IAA, and ethylene interacted with GAs and responded to the GAs treatment (Figure 23). There is a dynamic equilibrium relationship between GA_3_ and ABA in the presence of synergy and antagonism. In physiological and biochemical level of the study, it was found that the five hormones content all changed sharply by exogenous GA_3_ treatment, and the changes became stable after 4 h. From the ratio of the content of four hormones ABA, JA, IAA, and ZR to the content of GA_3_, the ratio within 4 h changed drastically and remained stable after 4 h, which indicated that 200mg/L exogenous GA_3_ treatment of the bulb for 4 h can quickly broke the hormonal balance and another equilibrium relationship was reached after 4 h. It was speculated that the phytohormone-related genes may express in a large amount within 4 h by exogenous GA_3_ treatment, and it also provided a reference for the subsequent determination of the processing time of the RNA-seq transcriptome sequencing test sample. Some studies have suggested that the ratio of ABA/GA_3_ is an important basis for determining whether plants are dormant or not. Following the changes in the external environment, plants regulate the dormancy and germination of seeds by self-regulating ABA/GA_3_. The cycle of this trend can be called the ‘sleep cycle’. Seeds can germinate when both internal gibberellin signals and external environmental signals meet the conditions [38]. In the process of breaking the dormancy of the bulb at low temperature, there is a balance of ABA/GA_3_ in the early stage of dormancy. During the period of sexual flowering, that is, during the period from 28 d to 35 d, ABA/GA_3_ decreased rapidly, and ABA/GA_3_ re-stabilized at a lower level from the 35th day after the completion of physiological flowering transition. In the process of treatment of lily bulbs with exogenous GA_3_, the changes of endogenous GA_3_ and ABA content were consistent with the trend of low temperature vernalization, and ABA/GA_3_ was treated by exogenous gibberellin at a higher level. Within 4 h, it quickly dropped to a lower level and remained steady. In summary, endogenous hormones may affect the differentiation of flower buds by affecting the metabolism of carbohydrates, and the ABA/GA_3_ ratio is decreased and re-stabilized at a lower level is an important indicator for completing the physiological flower-forming transformation. A study has found that the dormancy cycling in Arabidopsis seeds is controlled by seasonally distinct hormone-signaling pathway. The results were consistent with ABA signaling linked to deep dormancy in winter being repressed in spring concurrent with enhanced DELLA repression of germination as depth of dormancy decreased. Dormancy increased during winter as soil temperature declined and expression of ABA synthesis (*NCED6*) and gibberellic acid (GA) catabolism (*GA2ox2*) genes increased [39]. A research indicated that 14-3-3 proteins have been linked to ABA signaling, with its role in seed germination [40]. Additionally, gibberellins influence 14-3-3 protein accumulation, which implies that 14-3-3 proteins take part in cross-talk of the hormonal pathways of ABA and GA [41]. Another study indicated that *RGL2* links to the GA and ABA signaling pathway by influencing ABA biosynthesis and activation of *ABI5* and *ABI3* [42,43]. Someone proposed that the *AP2* domain plays a critical but cryptic role in the dual regulation of ABA and GA biogenesis in fine-tuning seed dormancy and germination [44].

Recent findings demonstrated that another phytohormone, auxin, is also critical for inducing and maintaining seed dormancy, and therefore might act as a key protector of seed dormancy. Emerging genetic data show that auxin protects and strictly regulates seed dormancy alongside ABA [45]. In our study, we found that plant hormones were involved in the gibberellin response process of lily as *ARF10*, IAA1 genes, and ethylene gene *AP2* expressed differently. Studies have found that IAA can affect the synthesis of gibberellin and also affect the signal transduction of gibberellin. Studies have shown that IAA levels in peas and tobacco can affect gibberellin activity levels [46]. Studies have shown that IAA can induce the expression of gibberellin synthesis genes *GA20ox* and *GA3ox*, and inhibit the expression of gibberellin antagonist *GA2ox* [47]. Studies have shown that IAA affects the synthesis of gibberellins by the degradation of the inhibitor Aux/IAA protein and the activation of the transcription factor *ARF7* [48]. Other studies have shown that the Arabidopsis auxin receptor *TIR1* affects the synthesis of gibberellin by mediating Aux/IAA degradation and ARF activation [47]. The above studies indicate that IAA has a role in gibberellin biosynthesis and *DELLA* degradation. In this study, it was found that the IAA content in the inner scales rapidly decreased to half of the original 2 h after the application of exogenous gibberellin, and the content rapidly increased during the 2 h to 4 h period. The content of gibberellin increased in 2 h, but it was far less than that in the course of 2–4 h. With the increase of gibberellin content, the IAA content decreased again from 4 h to 8 h, and the rate of increase of gibberellin content in the subsequent 4–8 h became very slow. With the increase of gibberellin content, the IAA content began to increase from 8 h to 16 h, and with the increase of auxin content, the increase rate of gibberellin increased from 8 h to 16 h, and the endogenous gibberellin content changed continuously, maintaining a dynamic balance with changes in endogenous IAA content. After the administration of exogenous gibberellin, the endogenous gibberellin content increased overall, while the endogenous IAA content decreased compared to the initial content. This indicates that IAA can promote the synthesis of gibberellin, and the IAA and gibberellin content changes always maintain a dynamic synergistic relationship.

In addition to ABA, GA, and auxin, nearly all other phytohormones are also likely involved in modulation of seed dormancy and germination, including ethylene (ET), brassinosteroids (BRs), jasmonic acid (JA), salicylic acid (SA), cytokinins (CTKs), and strigolactones (SLs). Previous studies showed that ET may affect seed germination through an ABA/GA-independent pathway [49]. However, whether and how ET affects GA biogenesis and signaling regarding seed dormancy and germination is largely unknown so far. The detailed mechanisms underlying the BR–GA crosstalk are elusive. SA is a plant hormone mainly associated with various defense pathways. Circumstantial evidence suggests that SA also regulates seed germination as a bifunctional modulator. SA inhibits germination by inhibiting the expression of GA-induced α-amylase genes under normal growth conditions [50]. SLs are a small class of carotenoid-derived compounds that regulate many aspects of plant development, through the signaling pathway with D53 (*DWARF 53*) as a repressor [51,52,53]. They also trigger seed germination in other species, evidently by reducing the ABA/GA ratio [54].

Studies have shown that *GASA* is a downstream target gene of *DELLA*, and gibberellin can promote the expression of *GASA*. Overexpression of the key proteins *Mn NINJAl* and *Mn NINJAs* in the jasmonic acid signal transduction pathway in Arabidopsis can affect the expression levels of the gibberellin-responsive genes *AtGASA1* and *AtGASA6* [55]. In this study, it was found that the content of jasmonic acid increased rapidly after treatment with exogenous gibberellin for 4 h, and the growth rate at 2 h to 4 h was significantly faster than that at 0 h to 2 h, which was consistent with the change trend of endogenous gibberellin content at 4 h. After the increase of endogenous gibberellin content slowed, the endogenous jasmonic acid content decreased. This also indicates that there are both synergistic and antagonistic changes between jasmonic acid and gibberellin. Studies have found that *KNOXI* protein induces the production of cytokinin to inhibit the synthesis of gibberellin and promote the inactivation of gibberellin. *SPY* promotes cytokinin response by inhibiting gibberellin signaling and regulates the balance between the two hormones [56,57]. It is indicated that *KNOXI* and *SPY* play an important role in regulating the balance between gibberellin and cytokinin. However, it is unclear how *SPY* promotes cytokinin response. In this study, the content of cytokinin decreased rapidly within 2 h after administration of exogenous gibberellin, and then slowly increased with the increase of endogenous gibberellin content, which was decreased compared with the initial content.

In summary, these plant hormones—including ET, BRs, JA, SA, CTKs, and SLs—regulate seed dormancy and germination, most likely by mediating the ABA/GA balance, although the interactions among these hormones and GA need further investigation, and some known detailed mechanisms are only the tip of the iceberg. Hormones and signaling compounds precisely regulate seed dormancy and germination through an integrated network of interactions with the ABA/GA_3_ balance as the central node.

### 3.3. Basal Metabolism, Secondary Messengers, and Histone Modifications Play Important Roles in GA-Responsive Process

In early 20th century, scientists proved that the nutritional status of plants could influence plant flowering through a lot of experiments. The results of the study showed that the gene expression in starch, carbohydrate metabolism, and nitrogen metabolism, were significantly upregulated during the process of GA treatment. Carbohydrates are thought to play a crucial role in the regulation of flowering, and *trehalose-6-phosphate* (*T6P*) has been suggested to function as a proxy for carbohydrate status in plants. A previous study showed that the loss of *TREHALOSE-6-PHOSPHATE SYNTHASE 1* (*TPS1*) causes *Arabidopsis thaliana* to flower extremely late, even under otherwise inductive environmental conditions. This suggests that *TPS1* is required for the timely initiation of flowering [58]. In our study, genes (contig_73276, Contig11405) encoding the *trehalose 6-phosphate phosphatase* have been found in the Module 1. *Soluble acid invertase* (*S-AIV*) has important biological functions relating to sucrose import [59] and sugar signals [60]. A previous research indicated that soluble acid invertase (EC 3.2.1.26) is thought to play a critical role in sucrose hydrolysis in muskmelon fruit [61]. In our study, genes (contig_68521) encoding the soluble acid invertase have also been found in the Module 1. *Poly [ADP-ribose] polymerase (PARPs*) encoding a plant ortholog of the human protein *Lys-Specific Demethylase1* (*LSD1*), is involved in *H3K4* demethylation, involved in positive regulation of flower development, inflorescence development, auxin biosynthetic process, cotyledon development, histone deacetylation, and oxidation-reduction process. *PARPs* work in coordination with *poly (ADP-ribose) glycohydrolases (PARGs)* to regulate transcriptional activity by the addition and removal of *ADP-ribose polymers*, respectively [62,63]. The results of a research showed that *RGL2 (DELLA)* undergoes GA-induced degradation, and this process is blocked by proteasome inhibitors and serine/threonine phosphatase inhibitors; however, serine/threonine kinase inhibitors had no detectable effect, suggesting that dephosphorylation of serine/threonine is probably a prerequisite for degradation of *RGL2 (DELLA)* via the proteasome pathway [64]. Our study provides evidence of the role of basal metabolism and histone modifications in GA-signaling pathway for LA hybrid Lily, which is necessary for the transition to reproductive growth. A new study demonstrated that *DOG1* mediates a conserved coat-dormancy mechanism including the temperature- and gibberellin (GA)-dependent pathways [65]. Subsequent studies suggested the importance of epigenetic regulation for *DOG1*.

### 3.4. WRKY, MYB, and CBF TFs Play Important Roles in GAs Response and Signal Transduction Process

Members of the *WRKY* transcription factor superfamily are essential for the regulation of many plant pathways. Three group I *WRKY* genes, *OsWRKY24, −53*, and *−70* showed additive antagonism of ABA and GA signaling. These results suggest that these *WRKY* proteins function as negative transcriptional regulators of GA and ABA signaling [66]. A recent study showed that *WRKY12* and *WRKY13* displayed negatively correlated expression profiles and function successively to regulate flowering in *Arabidopsis* thaliana. Molecular and genetic analyses demonstrated that *FRUITFULL* (*FUL*) is a direct downstream target gene of *WRKY12* and *WRKY13*. Interestingly, they also found that *DELLA* proteins *GIBBERELLIN INSENSITIVE (GAI)* and *RGA-LIKE1 (RGL1)* interacted with *WRKY12* and *WRKY13*, and their interactions interfered with the transcriptional activity of the *WRKY12* and *WRKY13*. Further studies suggested that *WRKY12* and *WRKY13* partly mediated the effect of GA3 on controlling flowering time [67]. *WRKY41* regulates Arabidopsis seed dormancy also through directly controlling *ABI3* transcription during seed maturation and germination [68]. Subsequent studies demonstrated that *MYB96*, the ABA-responsive *R2R3-type MYB* transcription factor, positively regulates seed dormancy and negatively regulates germination through mediating expression of *ABI4* and ABA biogenesis genes, including *NCED2* and *NCED6* [69]. In this study, significant *WRKY, MYB*, and *CBF* TFs were found differentially expressed during the exogenous GA_3_ treatment, which indicated that these *WRKY, MYB*, and *CBF* TFs play important roles in GAs response and signal transduction process.

### 3.5. GA Floral Induction Pathway in LA Lybrid Lily

Preliminary analysis of Module 1 concluded that the transcription factors *WRKY, WRKY7, NAC2, NAC11*, and *heat shock transcription factor A2* were continuously upregulated after 2 h of administration of exogenous gibberellin, and it was confirmed that these transcription factors could rapidly respond to gibberellin signals. It may also activate the related modification of phosphorylation, and is also involved in the metabolic regulation pathway of abscisic acid. The contig_85766 gene shown in orange in the figure below responds to organic nitrogen compounds, which are genes closely related to sugar metabolism. Previous studies have suggested that the C/N ratio is an important indicator for plants to break dormancy. In addition, a transcription factor *CBF*, which is closely related to glucose metabolism, was found to speculate that this transcription factor can also regulate the transduction metabolism of abscisic acid. The contig_136393 gene in the figure below is enriched into the function of transcriptional regulation, and it is speculated that this unannotated gene is also a key transcription factor. The gene interaction network map of Module 1 indicates that the transcriptional regulation of *WRKY* transcription factor hormone is more direct and can directly activate the phosphorylation-related modification. The transcription factor *NAC* regulates the phosphorylation-related phosphorylation modification by regulating the expression of transcription factor *WRKY*. The transcription factor *CBF* can directly regulate the genes involved in glucose metabolism without directly participating in the metabolic process of hormones by regulating *WRKY* transcription factors. It is speculated that the transcription factor of this module can respond rapidly to exogenous gibberellin signaling, and is also involved in the metabolic process of abscisic acid. It may be a network node where gibberellin and abscisic acid genes interact, and it can be determined that these transcription factors regulate the gibberellin pathway (Figure 24).

Preliminary analysis of Module 2 may be a related gene network for gibberellin signal transduction. The gibberellin-mediated signal transduction-related genes are enriched by various metabolic processes, redox processes, cation transport, and protein phosphorylation modification. The protein pathway-related genes indicate that the two diacylglycerol kinases found may be the intracellular second messenger receptors in the LA-based lily’s dazzling response to gibberellin signaling, and both diacylglycerol kinase genes are significantly upregulated. Several specific cation transport related genes in LA lily were also found, including *ammonium ion transporter AMT*, *plasma membrane ATPase*, and *calcium transport ATPase*. After treatment with exogenous gibberellin, the *ammonium ion transporter AMT* and *calcium transport ATPase* genes were upregulated and the *plasma membrane ATPase* gene was downregulated. Among the genes associated with histone phosphorylation modification, threonine/serine protein kinase, methyltransferase, and sucrose synthase are highly correlated. From the genetic map in Module 2, it can be speculated that the G-protein coupled receptor is a direct messenger receptor in the gibberellin-mediated signal transduction pathway, and the G-protein coupled receptor can activate the downstream ammonium ion transporter gene. The signal transduction is further expressed by the expression of *AMT* and the *calcium transporter ATPase* gene, in which the metabolism of various sugars and lipids provides an energy basis for the transduction of the gibberellin signal (Figure 25).

It is preliminarily determined that Module 3 is a cross-gene network node of the gibberellin pathway and the low temperature vernalization pathway. It was found that exogenous gibberellin treatment can activate the expression of genes involved in the low-temperature response pathway, as shown in the following gene interaction network diagram, the activation of the cold signal includes the dephosphorylation process, and is enriched in the function of low temperature response. Glycan endotransglucosylase hydrolase, indicating accumulation of soluble sugars during this process. This process is regulated by a variety of transcription factors *WRKY21, WRKY22, AP2-EREBP, EREB1, NAC1, cytochrome P450*, and *CYP81*. This suggests that this module may be a gibberellin-induced cryo-response-related module, which may be part of the key cross-gene network of the gibberellin pathway and the low temperature vernalization pathway. In the gene interaction network, it was found that the transcription factors *WRKY21, NAC1, EREB1, AP2-EREB1, cytochrome P450*, and *CYP81* may directly participate in the regulation of stress on low temperature signals, while the transcription factor *WRKY22* indirectly regulates the expression of *AP2-EREB1*. The whole regulation process activates the dephosphorylation reaction (Figure 26).

It is preliminarily determined that the gene function in Module 5 is related to hormone and low temperature response, and a specific transcription factor *AP2* related to hormone response is found, and this process is closely related to cellular protein modification and polysaccharide metabolism. All genes in this module were upregulated after 2 h of response to exogenous gibberellin and subsequently downregulated. Based on the correlation between genes, it is speculated that the unannotated gene speculates that it may be a serine/threonine protein kinase (Figure 27).

In the Module 12, three uncharacterized genes related to the flower-forming transformation were found to be upregulated rapidly after exogenous gibberellin treatment for 2 h, followed by downregulation at 4 h, and transient high expression appeared. In this process, the transcription factor *WRKY27* is involved in regulation. This module is functionally enriched in the L-phenylalanine metabolic pathway, and it is speculated that the exogenous gibberellin-mediated flowering transition may pass through the L-phenylalanine metabolic pathway.

Module 14 speculates that cell recognition of exogenous gibberellins and response to intracellular gibberellin signaling can activate phosphorylation of proteins, particularly protein phosphorylation of serine/threonine.

According to transcriptome analysis and gene co-expression network analysis, the molecular regulation mechanism of gibberellin-induced flower induction in LA lily ‘Aladdin’ was obtained (Figure 28). G-protein coupled receptor responding to the signal of exogenous GA_3_ activate several specific cation transport related genes encoding ammonium ion transporter AMT, plasma membrane ATPase, and calcium transport ATPase. The biosynthesis of endogenous gibberellin was activated, so extracellular gibberellin signals can be transferred to cells. From the very beginning, *GGPP* forms active GAs under the catalysis of a series of enzymes. Genes encoding *KO, GA20ox, GA2ox, GA3ox, P450*, etc. were found playing important roles in the process. *Ca2+/CAM* maybe the second messenger of gibberellin signal. Active GAs bind to the *GID1* receptor, which allows subsequent interaction between *GID1* and *DELLA* proteins to form the stable *GID1-GA-DELLA* complexes. Then the complexes are recognized by the *SLY1* based *SCFSLY1* complex, which ubiquitylates the *DELLA* proteins and causes their degradation by the *26S proteasome*. The biological process is actually phosphorylation and glycosylation of *DELLA* protein. Many transcription factors such as *WRKY7, WRKY21, WRKY22, NAC1, NAC2, NAC11, AP2, CBF, EREB1, CYP81*, etc. specifically regulate this biological process. The *DELLA* protein’s phosphorylation and dephosphorylation process is reversible depending on the regulation of different transcription factors as shown in the figure below. The polysaccharide metabolism exists for supplying energy for the process. The reduction of *DELLA* promotes the transcription of downstream factor *GASA* which promotes the expression of *SOC1(AGL20).* Subsequently, the *LFY* expressed a lot to stimulate flowering. A research indicated that GA-induced expression of *FT* is dependent on *CO* to modulate flowering under LDs [19]. In our study, *CO* also expressed differentially during the exogenous GA_3_ treatment. Furthermore, epigenetic and stress-resistant related genes and genes in the phytohormone metabolic pathway and L-phenylalanine metabolic pathway also participated the biological process.

## 4. Materials and Methods

### 4.1. Plant Materials and Exogenous GA3 Treatment

Bulbs of LA hybrid lily ‘Aladdin’ (Figure 29) cultivated at Beijing Forestry University greenhouse (116.3° E, 40.0° N) with growth condition as 70% relative humidity, 25 °C:18 °C/day: night temperatures were collected for experiments after the ground parts withered in October. The disease-free and pest-free bulbs with an even size of 16–18 cm circumference were selected, cleaned, disinfected (50% carbendazim WP 800 times aqueous solution for 30 min), dried, and then stored in moist sawdust at 25° C for four weeks to unify the physiological and biochemical condition of the dormancy [70]. Although rapid emergence can be induced by a short treatment (24 h) with gibberellins [71] in *L. speciosum* ‘Star Gazer’and ‘Snow Queen’ [8], gibberellin-treated bulblets do not grow after emergence [8]. Such a treatment thus induced dormancy breaking and floral stimulation in a rather unnatural way. The effect of gibberellin would be more significant if the hormone could be shown to be equivalent to the cold treatment rather than lead to an artificial, non-physiological reaction [72]. Therefore, bulbs were randomly divided into five groups and were respectively treated with a relatively high concentration 200mg/L [12,73,74] GA_3_ (Huayueyang Biotech, Beijing, China) for different times (0, 2, 4, 8, and 16 h). Sample applied in water without GA_3_ treatment (0 h) were used as control. Three independent replicates were performed. Shoot apical meristems (SAMs) at each of these time points were excised from the bulbs as the samples and immediately frozen in liquid nitrogen, stored at −80 °C for the later study of phytohormone and carbohydrates content assay, RNA extraction for RNA-seq transcriptome sequencing and RT-PCR. Roots, outer scales, and inner scales excised from the bulbs treated by exogenous GA_3_ in the way mentioned above were used in the study of carbohydrates content assay.

### 4.2. Phytohormone and Carbohydrates Content Assay

Five samples treated with exogenous GA_3_ mentioned above were used to determinate the phytohormone content by ELISA (enzyme-linked immunosorbent assays). Each sample includes 10 SAMs. Three independent biological replicates were performed.

Roots, outer scales, and inner scales processed as mentioned above were used to determinate the soluble sugar and starch contents by anthrone ethyl acetate. There is a total of 15 samples, including 3 tissue sites at 5 treatment times per tissue site. Each sample is 0.2g (dry sample). Three independent biological replicates were performed.

### 4.3. RNA Extraction and RNA-seq Library Construction

Total RNA was extracted from samples respectively treated by 200mg/L GA_3_ solution for 0 h(CK), 2 h, 4 h with an Easyspin Isolation System of RNAisomate RNA (Aidlab Biotech, Beijing, China) according to the manufacturer’s instructions. Sample applied in water without GA_3_ treatment (0h) were used as control. Each sample includes 10 SAMs. Three independent biological replicates were performed.

The yield of RNA was determined using a NanoDrop 2000 spectrophotometer (Thermo Scientific, USA), and the integrity was evaluated using agarose gel electrophoresis stained with ethidium bromide. The result showed that the total RNA is qualified (RIN ≥ 7.0, 28s/18s ≥ 0.7) (Table 6). The result of agarose gel electrophoresis are as follows (Figure 30), the integrity of RNA meets the demand for the following RNA-seq library construction.

Each total RNA extract was first treated with RNase-free DNase I (TaKaRa, Dalian, China) to remove contaminating DNA, and then concentrated the mRNA content by capturing on magnetic oligo (dT) beads, followed by reverse-transcription into first-strand cDNA using reverse transcriptase. Then second-strand cDNA was synthesized using RNase H, dNTPs, DNA polymerase I, and appropriate buffers with a SuperScript Double-Stranded cDNA Synthesis kit (Invitrogen, Camarillo, CA, USA). Then, cDNA was depurated and settled end repair with an elution buffer and by addition of poly (A). Sequencing adaptors were then ligated to the fragments, and agarose gel electrophoresis used to select the range of fragments suitable for PCR amplification. Sequencing using an Illumina HiSeq™2000 platform was performed at the ShoBiotechnology Corporation (SBC), Shanghai, China following the manufacturer’s protocols. Each sample was sequenced with three cDNA reactions independently as biological replicates.

### 4.4. De Novo Assembly and Functional Annotation

Using scaffolding contig methods with a minimum contig length of ≥400, pre-processing and de novo assembly were carried out including the removal of the adapter sequences, ambiguous inner regions, shorter-than-15-nucleotide sequences, and low quality (Q20, 20) sequences using the SOAP2 aligner.

Adaptor sequences, reads with unknown sequences and low-quality reads were first removed from data produced from sequencing machines, and the clean reads were then assembled using Trinity software [75] (http://trinityrnaseq.sourceforge.net). Data obtained for each sample were separately assembled, and the assembly sequences were called unigenes. The unigenes from all samples were further subjected to sequence splicing and redundancy removal with sequence clustering software to acquire non-redundant unigenes as long as possible. Some unigenes in which the similarity was more than 70% were designated by the prefix CL, the others were singletons, in which the prefix was unigene. These unigene sequences were aligned and annotated to protein databases like Nr, Swiss-Prot, COG, GO, KEGG, and nucleotide database Nt with a threshold of E < 10–5. The best aligning results were used to decide sequence direction of unigenes. If results from different databases conflicted with each other, apriority order of Nr, Swiss-Prot, KEGG, and COG should be followed when deciding sequence direction of unigenes.

### 4.5. Genes Differential Expression and Gene Functional Enrichment Analyses

RPKM method was used to calculate the expression. Using DEGseq, the difference in gene expression between samples were detected. Then according to the Audic & Claverie method, a rigorous algorithm was developed to identify genes expressed differentially. FDR was used to decide the threshold of P-value in tests and analysis. When log2 ratio ≥ 1 and FDR ≤ 0.05 between the samples, the genes were regarded as expressed differentially.

For unigene expression analysis, preprocessed RNA-seq reads were mapped to unique transcripts with Bowtie2, and then the unigene read counts were obtained by eXpress, which can correct multiple mapped reads. Then, differential express transcripts between two treatments without replicate were detected by R DESeq package, which normalized the library size based on library reads counts and detected the DE genes based on the negative binomial distribution. nbinomTest function was used for no-replicate analysis. Differential express transcript *p*-value was corrected by Benjamini and Hochberg FDR correction. Over-representation of GO terms of differential express unigenes were identified by BiNGO plugin in Cytoscape software with a hypergeometric test after Benjamini and Hochberg FDR correction at a significance level of *p*-value < 0.05 based on our custom tobacco transcripts GO annotated datasets. Over-representation of KEGG pathways of differential express unigenes were identified with GSEAKEGGHyperGParams function in R GOstats package, which was based on the hypergeometric test. *p*-values were corrected by Benjamini and Hochberg FDR method, then a significance level of *p* < 0.05 was selected. The KEGG native diagrams were obtained using keggview.native function in R Pathview package.

### 4.6. Gene Co-Expression Analysis of DEGs, Rank Calculation, and Construction of GCN

A DEG is identified if the associated FDR ≤ 0.05 and |log2 (ratio)| ≥ 1.5 were observed in any three pairwise transcriptome comparisons [76]. The heat-map of the total DEGs was generated using MeV4.9 and clustered by hierarchical clustering (HCL) with default parameters. Highest-reciprocal rank (HRR) based co-expression network methodology [77,78] was employed to further investigate the interactions between the DEGs to construct GCN. HRR are used as an index of gene co-expression and in the construction of the aforementioned gene co-expression networks. Pearson’s correlation coefficient (r) was used as a metric of similarity to define expression values between DEGs. Using the above method, correlation matrices were first calculated. Next, raw r values for every relationship between DEGs were transformed into HRR [77]. The HRR-based GCN was calculated using R version 3.3.1. (https://www.r-project.org/) with the parameter of HRR = 30, and then the GCN were visualized using Cytoscape [79]. A Cytoscape plugin, Cluster Maker [80], was used to detect the Markov cluster algorithm (MCL) [81] modules using inflation score (I) parameter of 1.6. HRR networks were generated using different cut-offs to identify functional clusters, where weights of 0.2, 0.067, and 0.04 were given cut-off HRR scores of 5, 15, and 25 respectively for performance evaluation. Visualization of gene co-expression network was performed using a combination of features introduced in Cytoscape 2.8 [79]. For network analysis, the Cytoscape plugin Network Analyzer [82] was used. Significantly enriched GO terms for each of the detected module was carried out by AgriGO Tool [83]. The Arabidopsis gene model *(TAIR9*) was chosen as the reference set, hypergeometric distribution adjusted by Hochberg false discovery rate (FDR) adjustment for the testing of multiple hypotheses with an adjusted threshold of *p* < 0.05. A minimum number of mapping entries of 2 were used to evaluate the statistical significance of the functional enrichment in detected modules.

### 4.7. Quantitative Real-Time PCR Validation

30 candidate genes were selected to determine the expression level using real time quantitative RT-PCR. According to the manufacturer’s specifications, total RNA was extracted from the five samples as mentioned above. The yield of RNA was determined using a NanoDrop 2000 spectrophotometer (Thermo Scientific, USA), and the integrity was evaluated using agarose gel electrophoresis stained with ethidium bromide.

Quantification was performed with a two-step reaction process: reverse transcription (RT) and PCR. Each RT reaction has two steps. The first step is 0.5 μg RNA, 2 μL of 4×gDNA wiper mix, add nuclease-free H2O to 8 μL. Reactions were performed in a GeneAmp^®^ PCR System 9700 (Applied Biosystems, USA) for 2 min at 42 °C. The second step is adding 2μL of 5 × HiScript II Q RT SuperMix IIa. Reactions were performed in a GeneAmp^®^ PCR System 9700 (Applied Biosystems, Foster City, CA, USA) for 10 min at 25 °C; 30 min at 50 °C; 5 min at 85 °C. The 10 μL RT reaction mix was then diluted × 10 in nuclease-free water and held at −20 °C. Real-time PCR was performed using LightCycler^®^ 480 II Real-time PCR Instrument (Roche, Basel, Switzerland) with 10 μL PCR reaction mixture that included 1 μL of cDNA, 5 μL of 2× QuantiFast^®^ SYBR^®^ Green PCR Master Mix (Qiagen, Hilden, Germany), 0.2 μL of forward primer, 0.2 μL of reverse primer, and 3.6 μL of nuclease-free water. Reactions were incubated in a 384-well optical plate (Roche, Switzerland) at 95 °C for 5 min, followed by 40 cycles of 95 °C for 10 s and 60 °C for 30 s. Each sample was run in triplicate for analysis. At the end of the PCR cycles, melting curve analysis was performed to validate the specific generation of the expected PCR product. The primer sequences were designed by Beacon Designer and synthesized by Generay Biotech (Generay, Shanghai, China) based on the mRNA sequences obtained from transcriptome database as following table (Table 7 and Table 8) and the mRNA sequences obtained from DEGs for co-expression analysis. All experiments were operated in biological triplicates. The expression levels of mRNAs from transcriptome database were normalized to the internal reference gene *LoTIP1* (F: CGAAGCCAGAAACGGAGAAGAAT, R: GGGTAGGGTGGATTGGGAAGA) *18S* (F: CGGCTACCACATCCAAGGAA, R: GCTGGAATTACCGCGGCT). The expression levels of mRNA from DEGs for co-expression analysis were normalized to the internal reference gene *LoTIP1* (as above) and *GAPDH* (F: CACGGTCAGTGGAAGCACCATGAGAT, R: AGCAGCAGCCTTATCCTTGTCAGTGA). All were calculated using the 2-ΔΔCt method [84].

## 5. Conclusions

200 mg/L exogenous GA3 can successfully break the bulb’s dormancy of the LA hybrid lily and significantly accelerated the flowering process, indicating that gibberellin floral induction pathway is present in the LA lily ‘Aladdin’. With the GCN analysis, two second messenger G protein-coupled receptor related genes that respond to gibberellin signals in the cell were discovered. The downstream transport proteins such as *AMT*, calcium transport *ATPase*, and plasma membrane *ATPase* were also found to participate in GA signal transduction. Transcription factors—including *WRKY7, NAC2, NAC11*, and *CBF*—specially regulated phosphorylation and glycosylation during the ubiquitination degradation process of DELLA proteins. These transcription factors also activated in abscisic acid metabolism. A large number of transcription factors—such as *WRKY21, WRKY22, NAC1, AP2, EREB1, P450*, and *CYP81*—that both regulate gibberellin signaling and low-temperature signals have also been found. The specific molecular regulation mechanism of GA response and transduction to release dormancy and promote germination is still unknown, but an extensive range of putative mechanisms of GA floral induction pathways in the LA hybrid lily ‘Aladdin’ have been constructed.

## Figures and Tables

**Figure 1 ijms-20-02694-f001:**
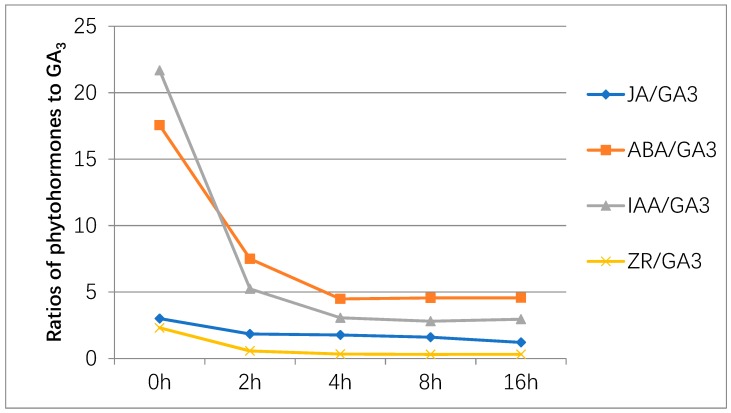
Changes of the ratios between four endogenous phytohormones contents to endogenous gibberellins contents with the exogenous GA_3_ treatments for 0, 2, 4, 8, and 16 h.

**Figure 2 ijms-20-02694-f002:**
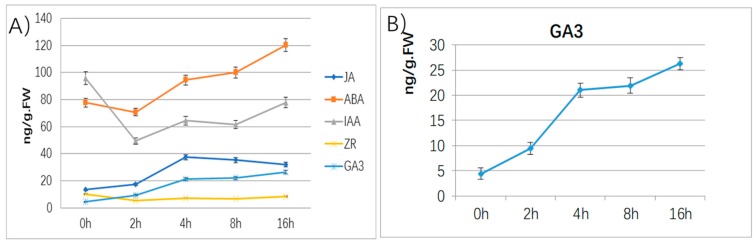
Changes of endogenous phytohormones contents during exogenous GA_3_ treatment. (**A**) Comparison of the contents of five endogenous phytohormones during exogenous GA_3_ treatment. (**B**) Changes of gibberellins contents during exogenous GA_3_ treatment. (**C**) Changes of ABA contents during exogenous GA_3_ treatment. (**D**) Changes of JA contents during exogenous GA_3_ treatment. (**E**) Changes of IAA content during exogenous GA_3_ treatment. (**F**) Changes of ZR content during exogenous GA_3_ treatment.

**Figure 3 ijms-20-02694-f003:**
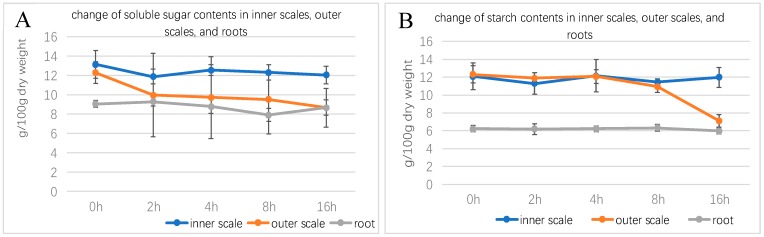
Changes of soluble sugar and starch contents in different tissue parts during the exogenous GA_3_ treatment. (**A**) Comparison of soluble sugar contents in inner scales, outer scales, and roots. (**B**) Comparison of starch contents in inner scales, outer scales, and roots.

**Figure 4 ijms-20-02694-f004:**
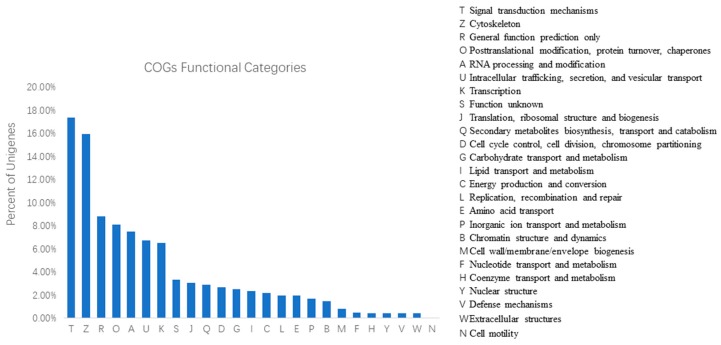
Clusters of orthologous groups (COG) classifications.

**Figure 5 ijms-20-02694-f005:**
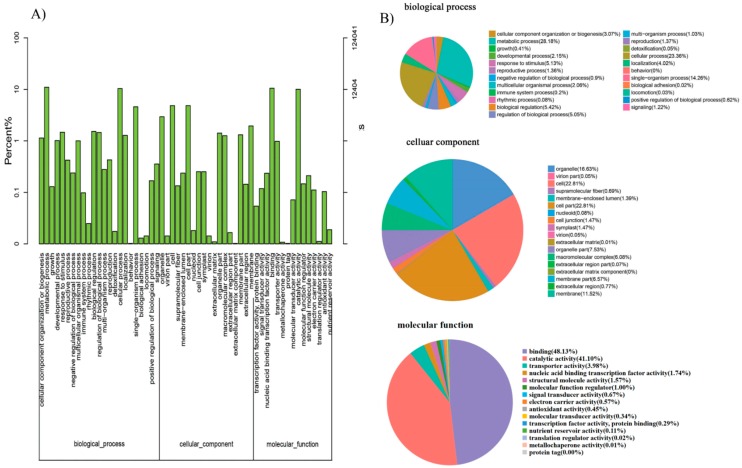
GO classifications and GO enrichment of LA hybrid lily transcriptome sequences. (**A**) Classification of GO. (**B**) Three main GO classifications and the representative terms.

**Figure 6 ijms-20-02694-f006:**
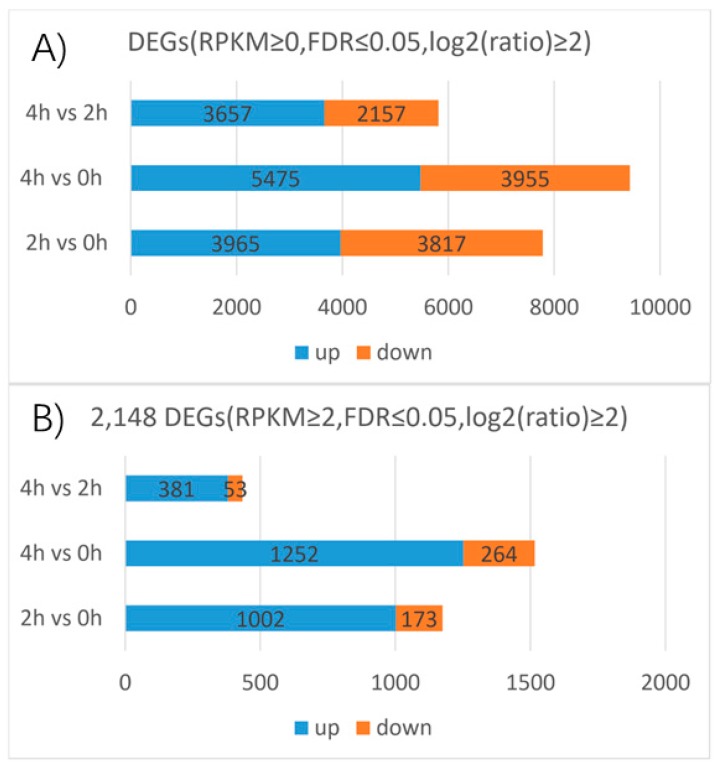
Up and down differential expression of genes. (**A**) DEGs were filtered with an RPKM ≥ 0, FDR ≤ 0.05 and |log_2_(ratio)| ≥2. (**B**) DEGs were filtered with an RPKM ≥ 2, FDR ≤ 0.05 and |log_2_(ratio)| ≥ 2.C).

**Figure 7 ijms-20-02694-f007:**
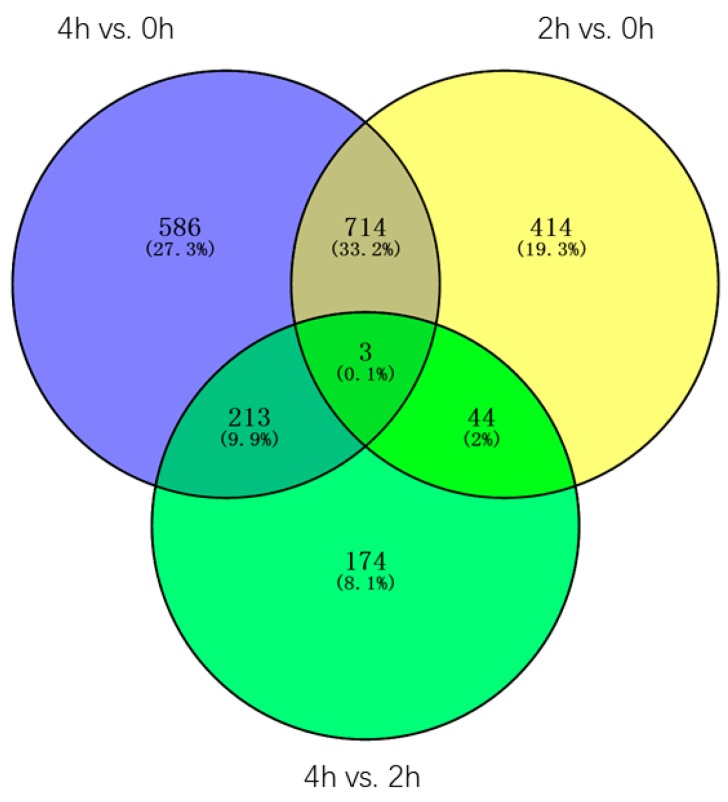
Venn diagram of 2148 DEGs expression profile changes.

**Figure 8 ijms-20-02694-f008:**
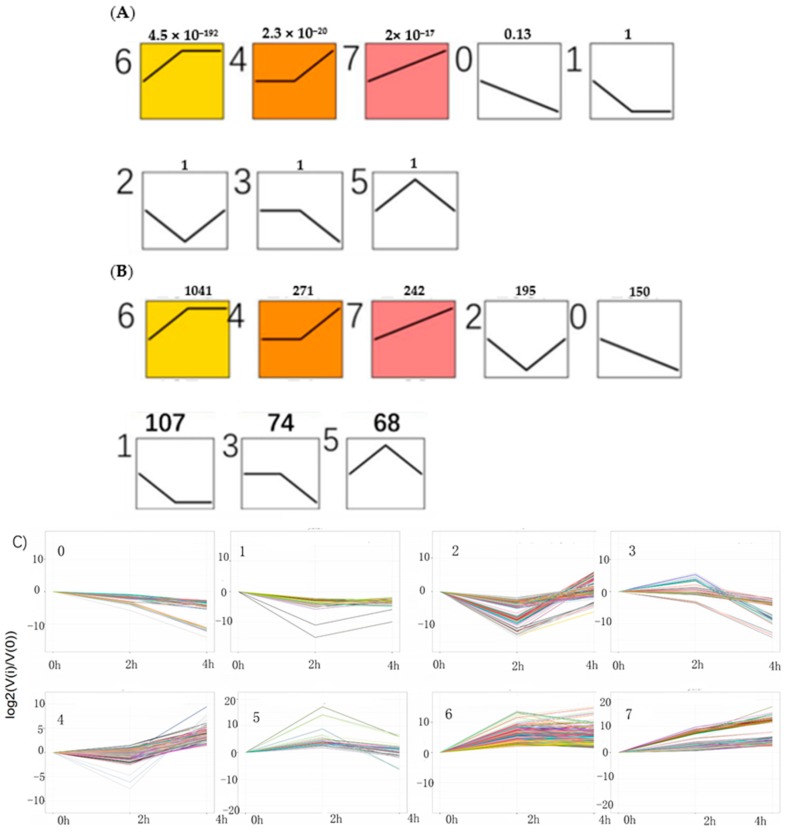
Expression trends of the total differentially expressed genes (DEGs). The profile’s number is placed on the upper left corner of each graph. (**A**) P-value of each profile from low to high. (**B**) Gene number of each profile from high to low. (**C**) Genes with the same expression trends are placed in the same profile.

**Figure 9 ijms-20-02694-f009:**
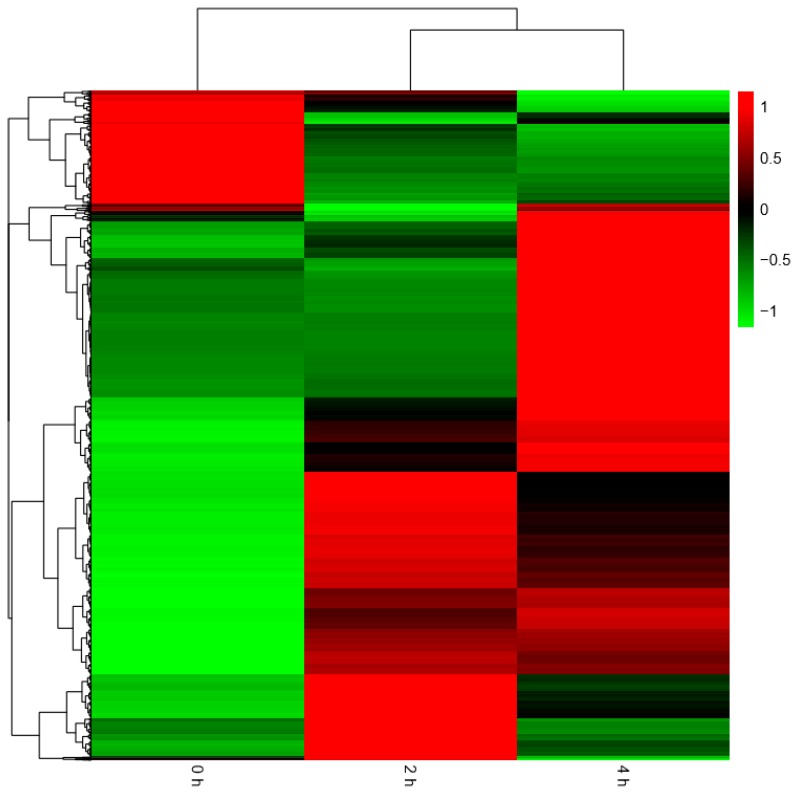
Heat-map of the total DEGs.

**Figure 10 ijms-20-02694-f010:**
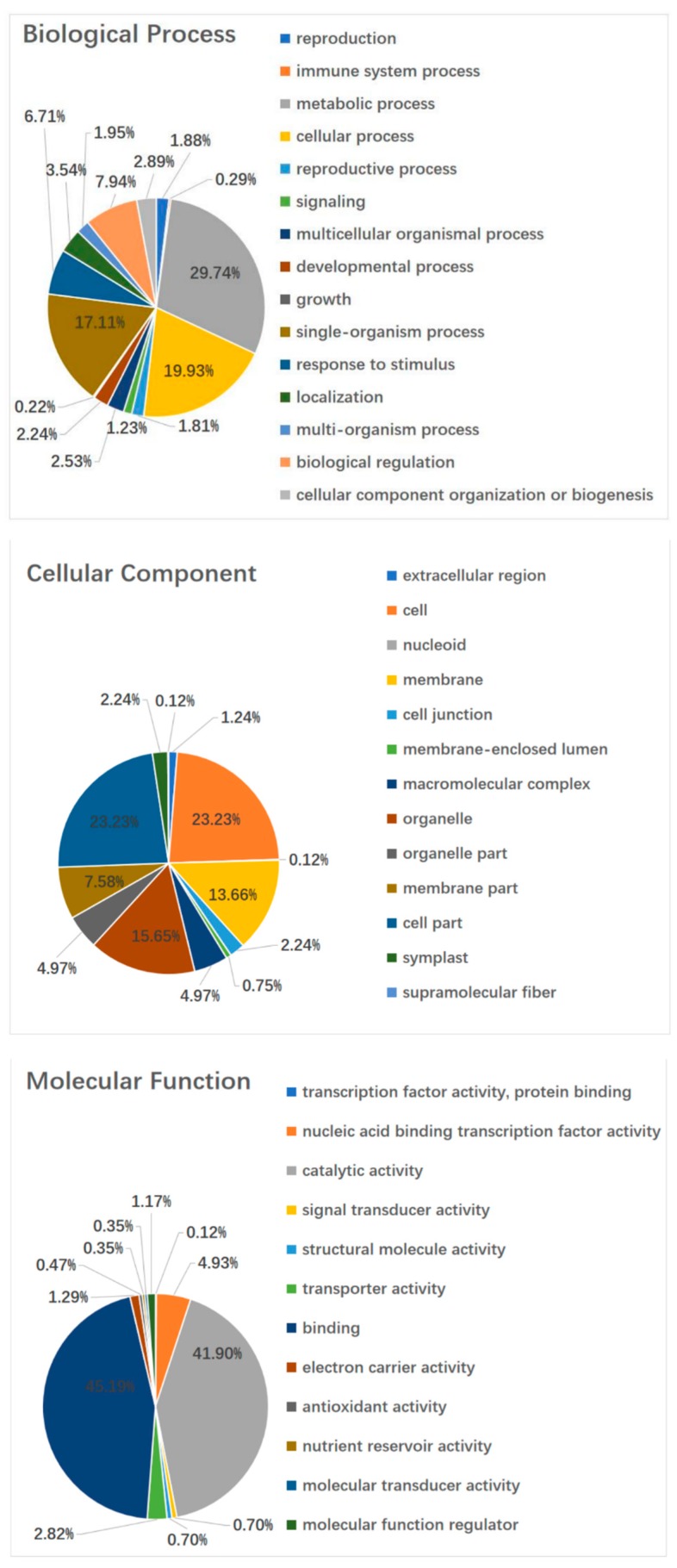
GO enrichment and GO classifications of lily genes.

**Figure 11 ijms-20-02694-f011:**
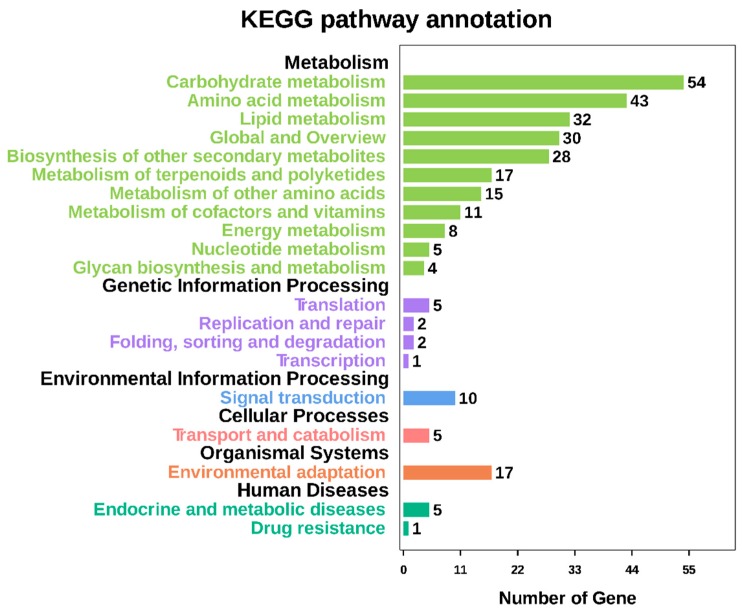
Classification of KEGG (2148 DEGs).

**Figure 12 ijms-20-02694-f012:**
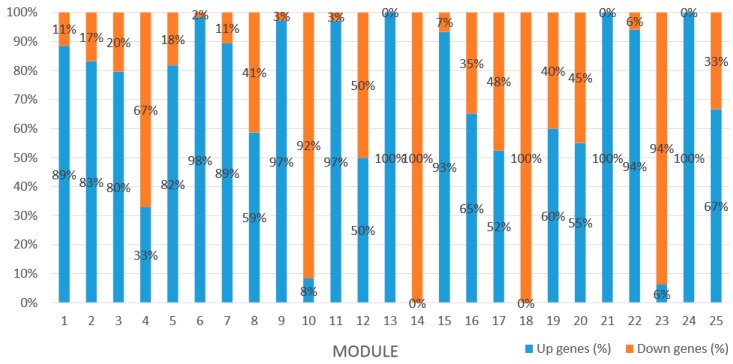
Expression trends of genes in modules.

**Figure 13 ijms-20-02694-f013:**
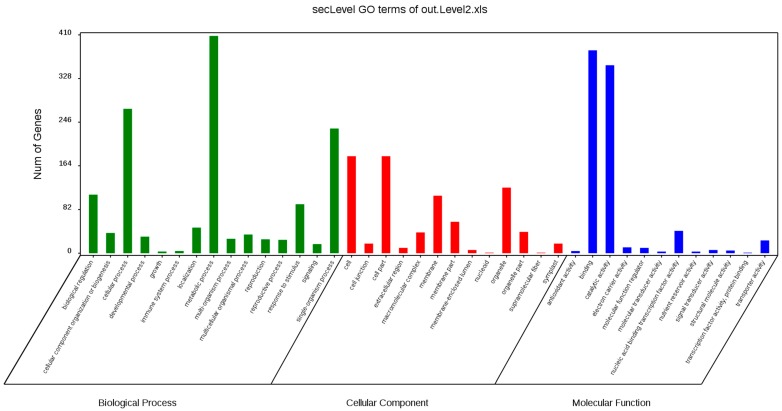
Classification of GO of GCN.

**Figure 14 ijms-20-02694-f014:**
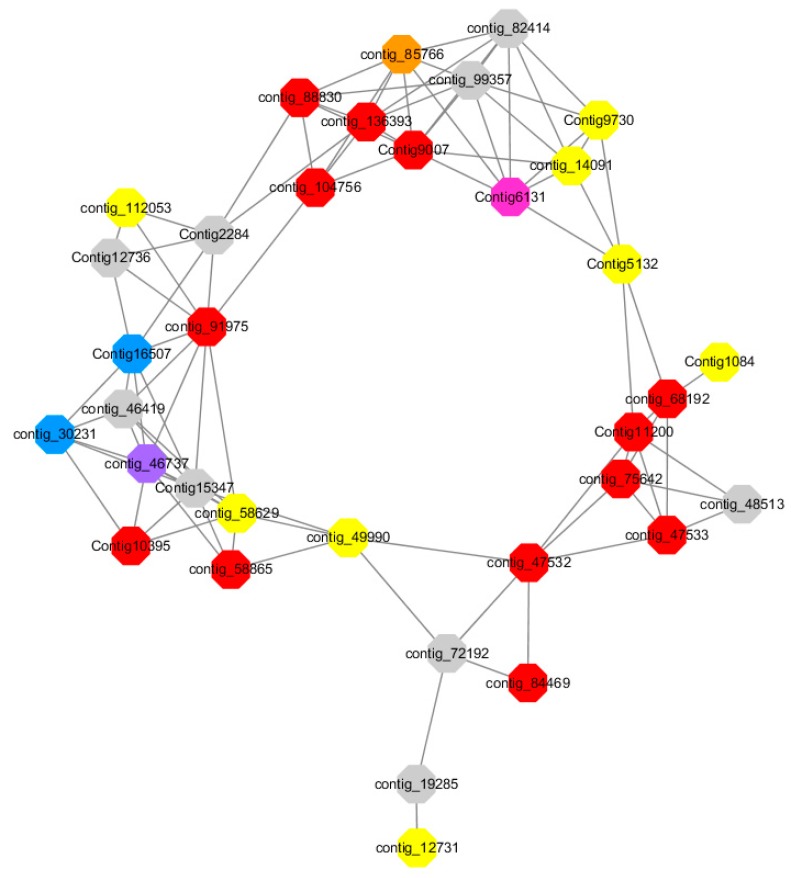
GCN of Module 1.

**Figure 15 ijms-20-02694-f015:**
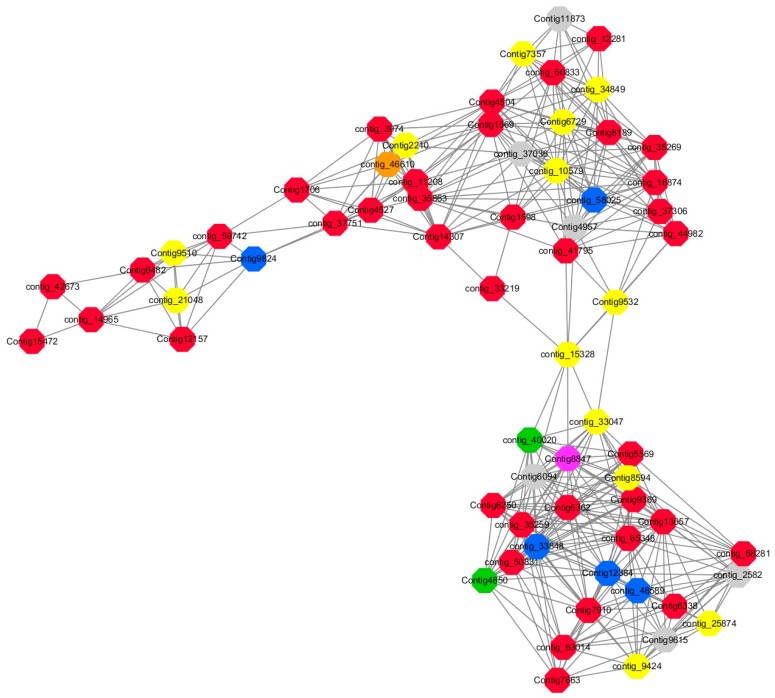
GCN of Module 2.

**Figure 16 ijms-20-02694-f016:**
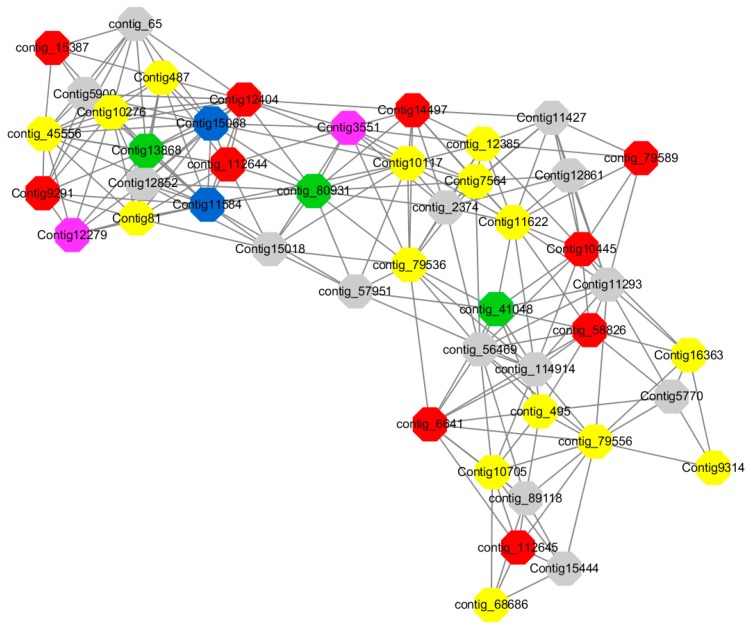
GCN of Module 3.

**Figure 17 ijms-20-02694-f017:**
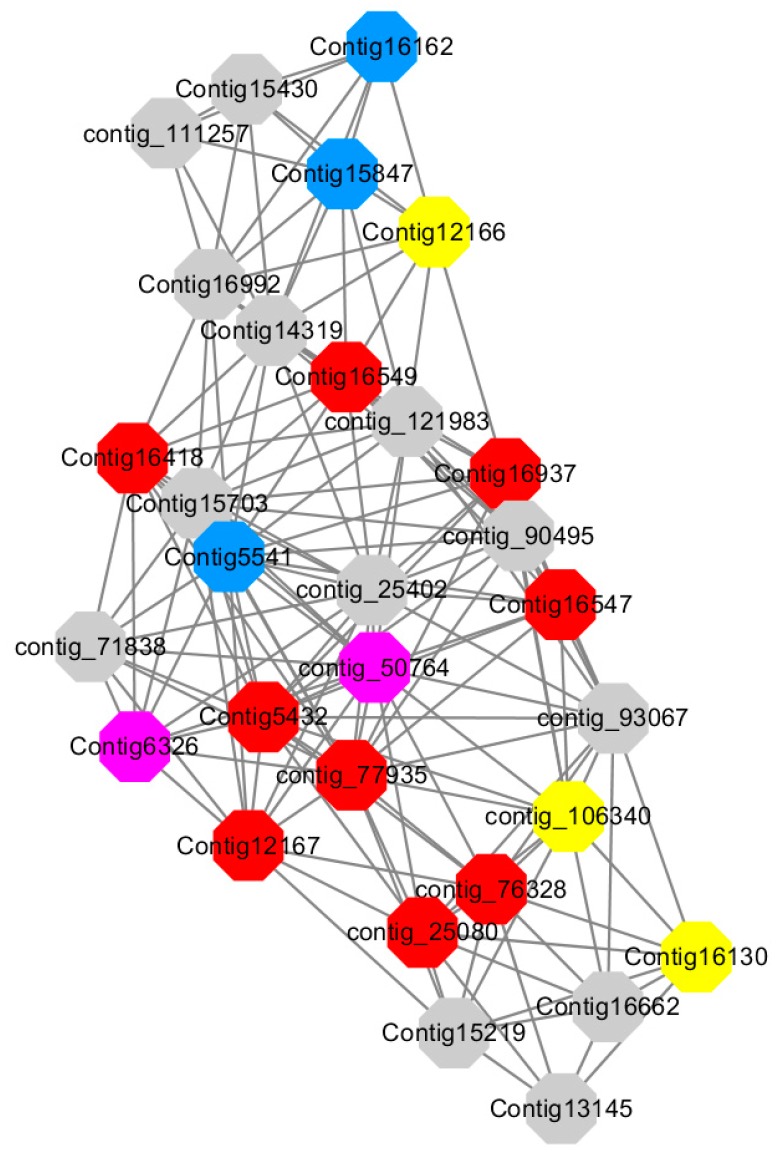
GCN of Module 5.

**Figure 18 ijms-20-02694-f018:**
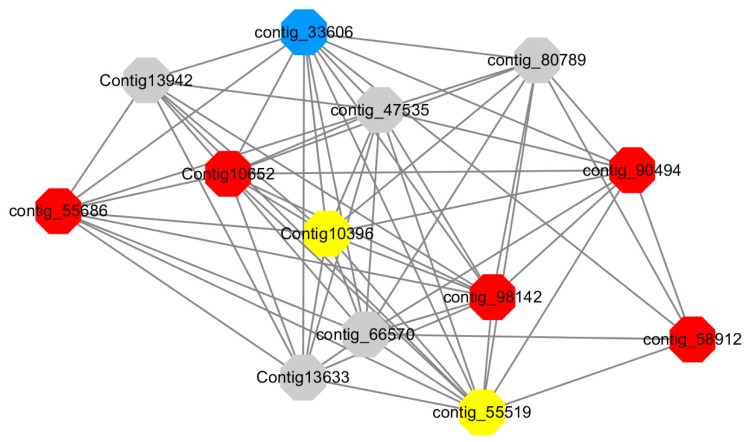
GCN of Module 12.

**Figure 19 ijms-20-02694-f019:**
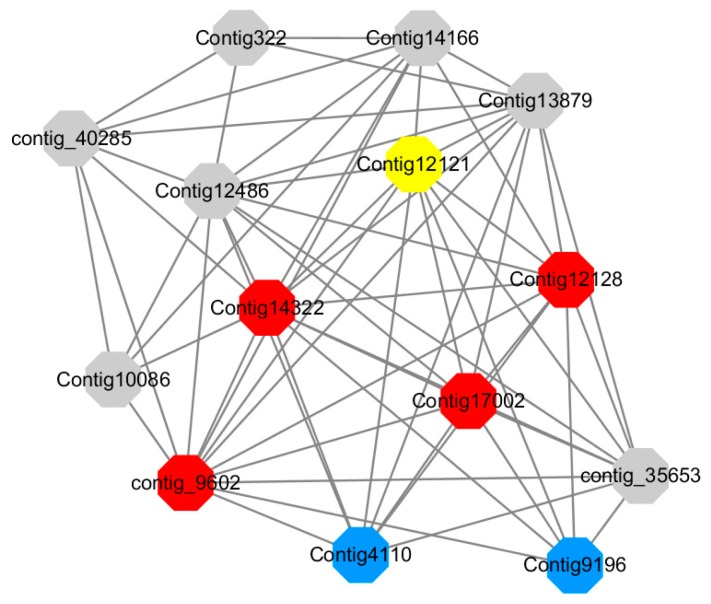
GCN of Module 14.

**Figure 20 ijms-20-02694-f020:**
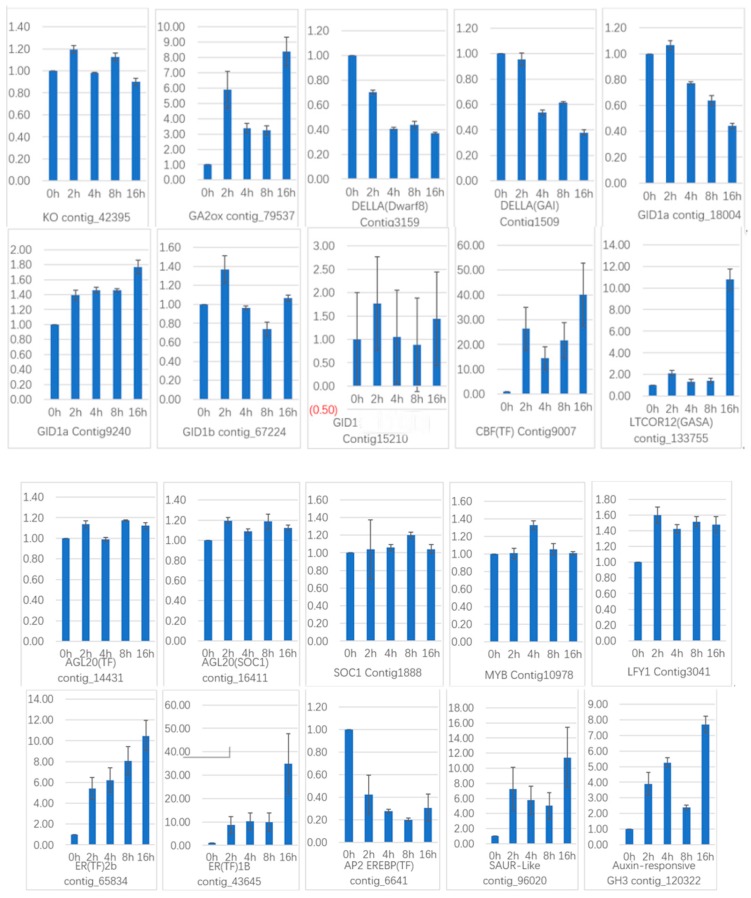
qRT-PCR of 20 key genes selected from transcriptome data.

**Figure 21 ijms-20-02694-f021:**
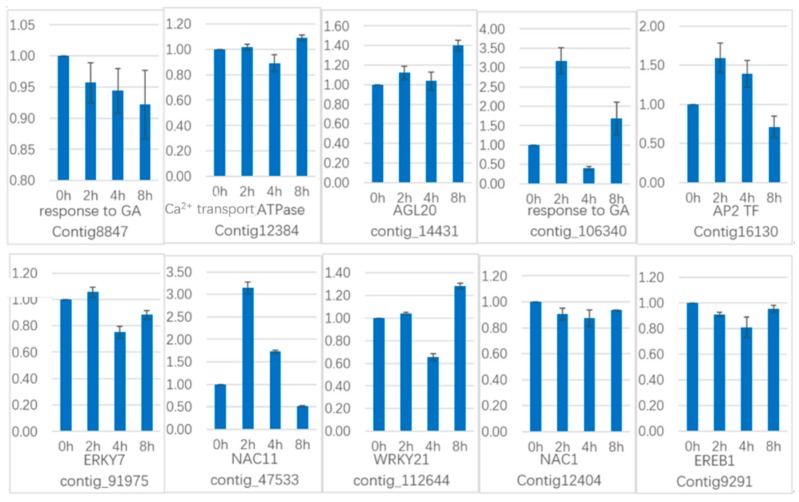
qRT-PCR of 10 key genes in GCN modules.

**Figure 22 ijms-20-02694-f022:**
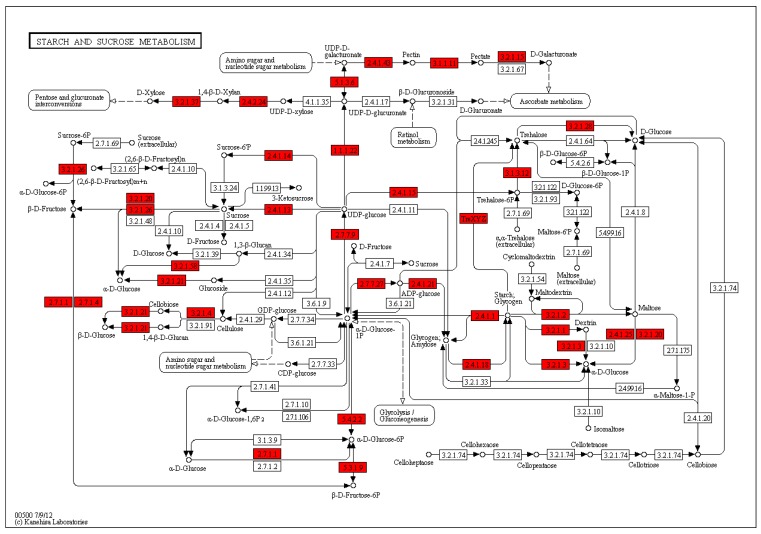
Starch and sucrose metabolism pathway. The red boxes indicate Unigenes were assigned to the enzymes.

**Figure 23 ijms-20-02694-f023:**
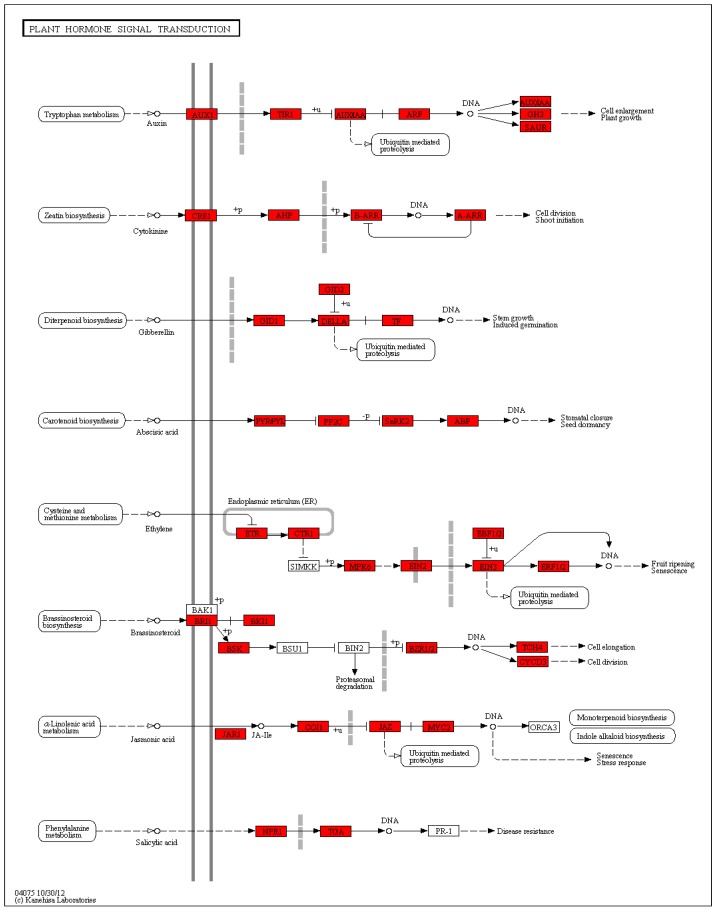
Plant hormone signal transduction pathway. The red boxes indicate Unigenes were assigned to the metabolic pathways.

**Figure 24 ijms-20-02694-f024:**
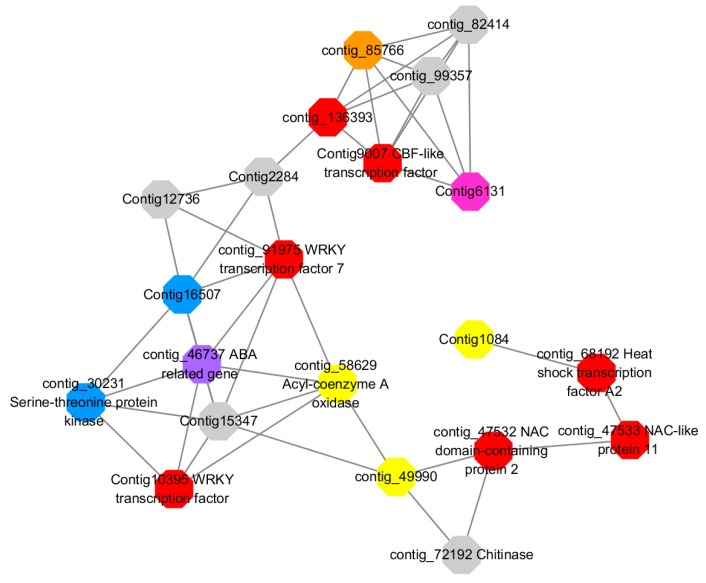
Interaction between genes in Module 1.

**Figure 25 ijms-20-02694-f025:**
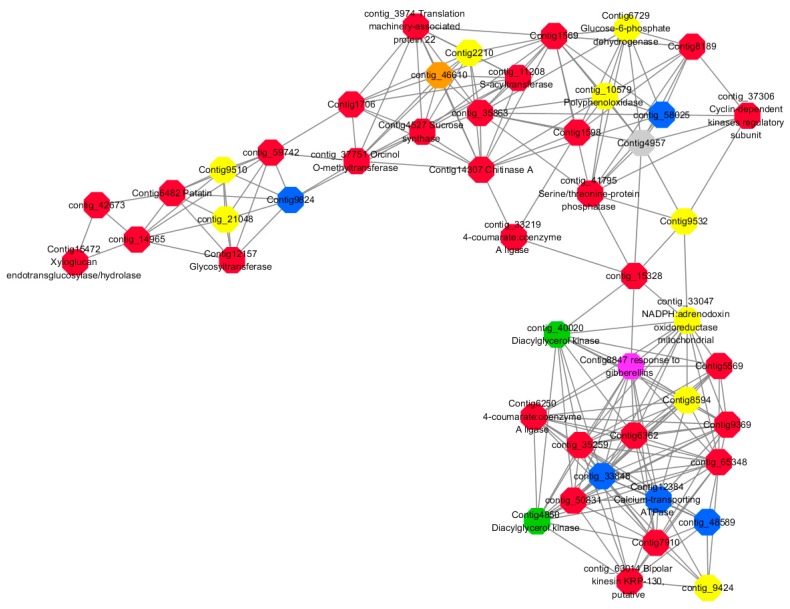
Interaction between genes in Module 2.

**Figure 26 ijms-20-02694-f026:**
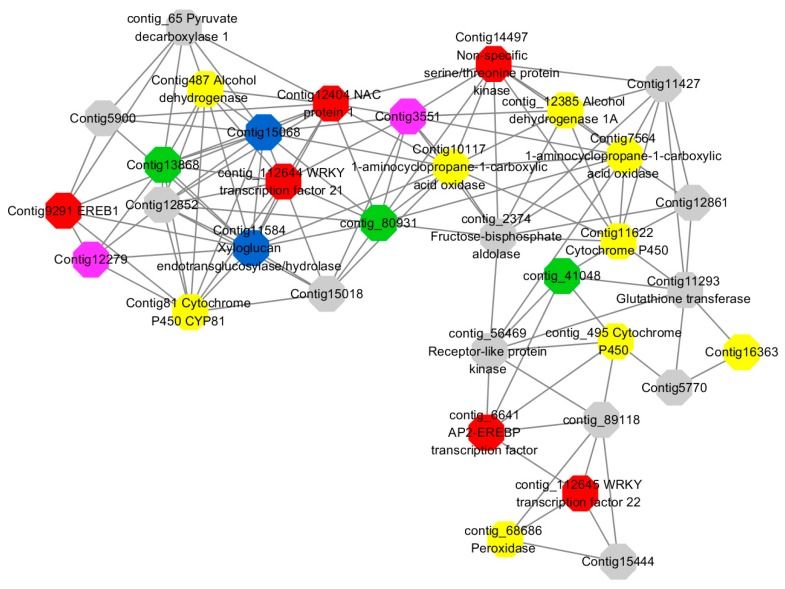
Interaction between genes in Module 3.

**Figure 27 ijms-20-02694-f027:**
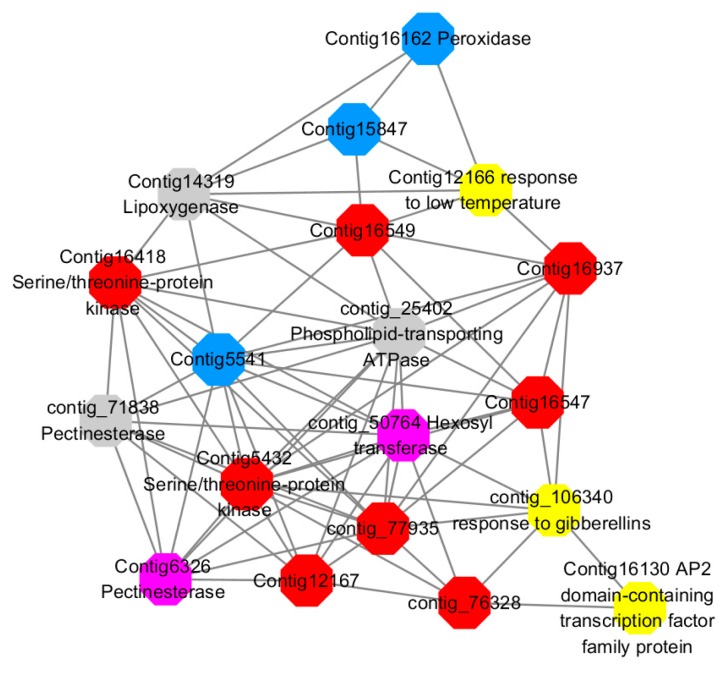
Interaction between genes in Module 5.

**Figure 28 ijms-20-02694-f028:**
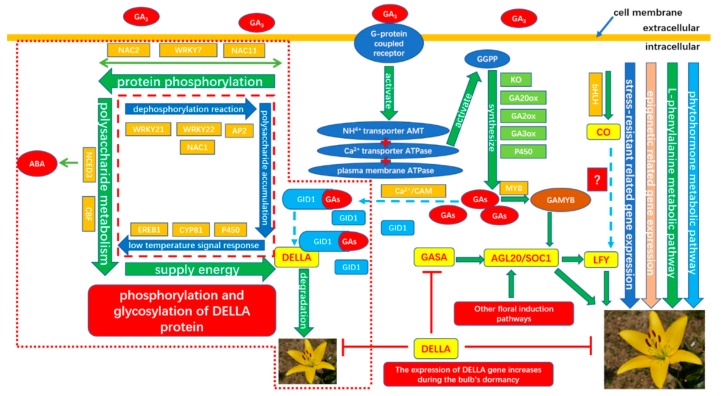
The molecular mechanism of floral transition induced by exogenous GA_3_ in LA hybrid lily ‘Aladdin’.

**Figure 29 ijms-20-02694-f029:**
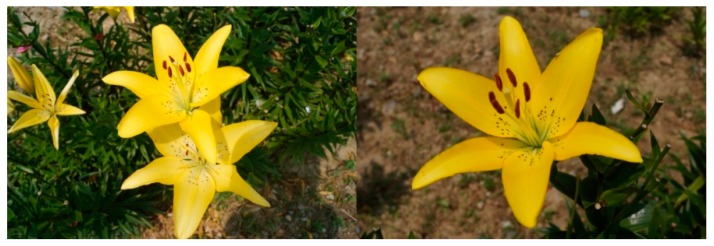
Experimental material.

**Figure 30 ijms-20-02694-f030:**
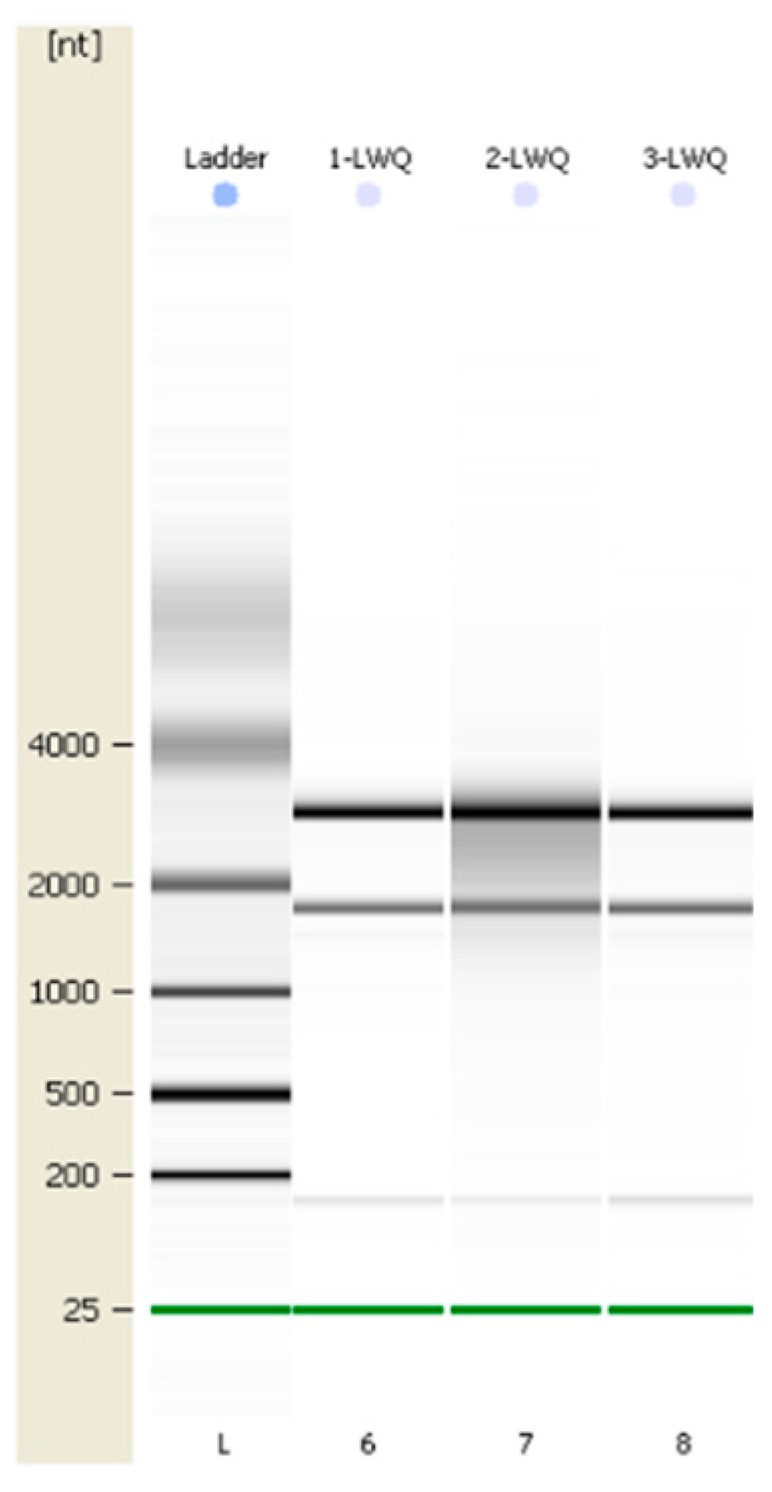
RNA integrity.

**Table 1 ijms-20-02694-t001:** Summary of sequencing and assembly data.

Sample ID	Raw Reads (MB)	Raw Bases (GB)	Q20 Value (%)	Raw Reads	Adaptor Trimed	Clean Ratio	rRNA Trimed	rRNA Ratio
0h	47.1	7.06	96.86	47,069,982	43,801,942	93.06	43,730,251	0.16
2h	54.3	8.15	96.40	54,301,760	49,605,630	91.35	49,509,627	0.19
4h	49.8	7.47	96.55	49,810,350	45,767,482	91.88	45,691,495	0.16

**Table 2 ijms-20-02694-t002:** Summary for the Oriental hybrid lily ‘Sorbonne’ transcriptome.

Statistics	Counts	Total Length (bp)	N25 (bp)	N50 (bp)	N75 (bp)	Average Length	Longest (bp)	N%	GC%	Annotation Count sNR/Uniprot	Annotation Ratio% NR/Uniprot
Contigs	168,599	87,457,055	1151	616	365	519	11,473	0.8	44.5		
Primary UniGene	153,850	88,130,297	1290	695	395	573	12,654	0.8	44.5		
Final UniGene	124,041	74,396,629	1432	756	408	599.77	12,655	0.9	43.87	48,927/48,725	39.44/39.28

**Table 3 ijms-20-02694-t003:** Annotation results.

	Number of Total Sequences	Number of Comments	Number of Notes
NR	124,041	48,927	39.44%
Uniprot	124,041	48,725	39.28%

**Table 4 ijms-20-02694-t004:** Classification of COG.

Code	Function	Unigene Number	Frequency
T	Signal transduction mechanisms	17,076	17.36%
Z	Cytoskeleton	15,655	15.92%
R	General function prediction only	8685	8.83%
O	Posttranslational modification, protein turnover, chaperones	7989	8.12%
A	RNA processing and modification	7381	7.50%
U	Intracellular trafficking, secretion, and vesicular transport	6637	6.75%
K	Transcription	6394	6.50%
S	Function unknown	3270	3.32%
J	Translation, ribosomal structure and biogenesis	3026	3.08%
Q	Secondary metabolites biosynthesis, transport and catabolism	2832	2.88%
D	Cell cycle control, cell division, chromosome partitioning	2637	2.68%
G	Carbohydrate transport and metabolism	2472	2.51%
I	Lipid transport and metabolism	2330	2.37%
C	Energy production and conversion	2159	2.19%
L	Replication, recombination and repair	1931	1.96%
E	Amino acid transport	1925	1.96%
P	Inorganic ion transport and metabolism	1663	1.69%
B	Chromatin structure and dynamics	1423	1.45%
M	Cell wall/membrane/envelope biogenesis	786	0.80%
F	Nucleotide transport and metabolism	487	0.50%
H	Coenzyme transport and metabolism	403	0.41%
Y	Nuclear structure	396	0.40%
V	Defense mechanisms	393	0.40%
W	Extracellular structures	393	0.40%
N	Cell motility	23	0.02%

**Table 5 ijms-20-02694-t005:** Parameters of GCN and HRR between co-expression genes.

Simple Parameters			
Clustering coefficient	0.758	Avg. number of neighbours	22.752
Connected components	49	Number of nodes	1296
Network diameter	7	Network density	0.018
Network radius	1	Network heterogeneity	0.448
Network centralization	0.024	Isolated nodes	19
Shortest paths	117878 (7%)	Number of self-loops	0
Characterislic path length	2.528	Multi-edge node pairs	0

**Table 6 ijms-20-02694-t006:** RNA quality.

No.	Sample Name	Con. (ng/μL)	Vol. (μL)	Total (μg)	A260/A280	2100 Result	
						RIN	28S/18S
1	0h	1896	55	104.28	2.18	10	1.9
2	2h	268	55	14.74	2.05	8.9	2.2
3	4h	1620	55	89.1	2.16	9.8	1.8

**Table 7 ijms-20-02694-t007:** Primers used in real-time quantitative PCR (RT-qPCR).

No.	Gene Symbol	Forward Primer	Reverse Primer	Product Length (bp)	Ta (°C)
1	contig_42395	GCTGATTTCTACAACGGCT	TGGAGTTTGGTGGTGGTAAG	111	60
2	contig_79537	CTCAACCACTACCCTTCCT	TCAGAGCAATCTGATACCCG	124	60
3	Contig3159	GCCATCCAACGATCAGTG	CCAAAGGGTAAATTCCCAGC	129	60
4	Contig1509	AGGTAGACCTCCGACATC	GAGTCGCTCCACTACTATTCC	101	60
5	contig_18004	GTATTCGGTTCCCAAAGAGC	TACACAAGATCGACATCCTGC	104	60
6	Contig9240	GCTACTAAACTCGACGGTCT	ATCATGTCCGGCTTTCTCAA	117	60
7	contig_67224	GAACAAGCGACGATTGGATTC	AGTGTAAATCCTAGCAATTGGT	106	60
8	Contig15210	AATGTGGATGTCGGACTTG	GGGAAGATAAGAAGTTCCGTG	101	60
9	Contig9007	CGAACCAAATTCCGCGAAA	TCGACTTCTTGTTGGGCT	103	60
10	contig_133755	CAGTCTCAGAGACAGCGA	TTGCCTCCTCATCTCATCAA	107	60
11	contig_14431	ATCAACATCAACAGGCCTACA	ACCCATCACCTTCCGTTTA	111	60
12	contig_16411	ACATCAACAGGCCTTCAGA	ACCCATCACCTTCCGTTTA	107	60
13	Contig1888	GTTGGAGAAGAATGCATCACT	CCTCTCAATAAACAGCTCGG	121	60
14	Contig10978	CAAAGGGTGGACTGATGC	AGATGAGAAGATGTTGATTGGC	101	60
15	Contig3041	CAGCCCATTCTTCTTCGC	GAGTCAGGCTCGTCGGAAA	108	60
16	contig_65834	ATGCGATCCTTCGGATTG	GATGGACCATCTCAGACTTCA	118	60
17	contig_43645	CCATCGAATCCTTGCCACTA	GTTGTACCAGTGAAGGTGC	116	60
18	contig_6641	GACTCATCAACGTCGTAGACA	TTCGTGGACCTGATGCTTT	100	60
19	contig_96020	TCGGTTCGATCATGTGAAGAC	ATCTGTGGGTAACCCAGCA	101	60
20	contig_120322	TTCTACAGCCTCCTAATGCC	GGTGGATTCATGTTGACTACC	116	60
21	TIP1	CGAAGCCAGAAACGGAGAAGAAT	GGGTAGGGTGGATTGGGAAGA	192	60
22	18S	CGGCTACCACATCCAAGGAA	GCTGGAATTACCGCGGCT	187	60

**Table 8 ijms-20-02694-t008:** Primers used in real-time quantitative PCR (RT-qPCR).

Gene Symbol	Forward Primer	Reverse Primer	Product Length (bp)	Ta (°C)
Contig8847	ACAATGAAAGGAGTCACTAACA	TTAGAAGCTCTTCAGAGTGCC	60	60
Contig12384	CCGAGTAGTGAAGGGAGTT	CTAGGAGTCCGATTGGCT	65	60
contig_14431	ATATCAACATCAACAGGCCTAC	ATTCAATCTTCTTCGTGAGGTC	81	60
contig_106340	TTGATGATAGCTTCTGGTGGG	AGTCCTCCATCGAACTCGTAA	60	60
Contig16130	TCAGAGTGCGATGAGTACGA	CCGCATTGGTCAGTTAAGTAT	91	60
contig_91975	ATTTGGATCTCTATGCTCACG	TCTCCCTCTCTTTATCTCCTGT	84	60
contig_47533	GACTCTTTGTACGATCCAACG	GATAAGGGAGAGACGGCAA	62	60
contig_112644	TAGACGGTGTTGACGGGA	GAGAGGTTTGCACGTTAGG	86	60
Contig12404	GGATTACAAATAGTAGCGGAGC	AGATTCAAGTCCATGAACCAC	65	60
Contig9291	GTTCAGATCGAACGGCAG	GACGCCGACGTTGTATACCA	60	60
TIP1(KJ543466)	CGAAGCCAGAAACGGAGAAGAAT	GGGTAGGGTGGATTGGGAAGA	192	60
GAPDH(KJ543468)	CACGGTCAGTGGAAGCACCATGAGAT	AGCAGCAGCCTTATCCTTGTCAGTGA	180	60

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
