# Peer review of "Transcriptional Regulatory Network of GA Floral Induction Pathway in LA Hybrid Lily"

_ijms, 2019, doi:10.3390/ijms20112694_

Round 1
Reviewer 1 Report
Introduction should be rewritten in more ordered way - at present the purpose of the study was defined in different paragraphs. Possibly introduction can also be reduced. Explanation of abbreviations i.e. gene names used in manusript optionally can make an article more friendly to readers. List of abbreviations should be extended (FW, ZR etc)
Fig 1 - improve legend, Fig 3 - part A+B+C is compiled and repeated in part D+E, and possibly one part will be sufficient,
In methods section evaluation of RNA quality should be supplemented.
Numbers need correction (table - line 203, fig line 220, 243), check number of sequences (line 212). Fig 4 and table 4 have duplicated contens and frequencies not fit - so it should be checked and better to leave table and remove Fig.4. Fig 5. Fig 6 (right side not desribed) is hard to read - not clear. Table 5 - maybe better to present full lenght in supplemntary materials. To improve clarity of presentation all names of contigs should be shortened and reduced to necessary unique abreviation (also in figs). Sequences of contigs mentioned in publication and ideally all should be deposited in some public database. Source of transcripts in Fig 7 should be clearly explain in caption. Fig.8 - seems that left part of fragmnent A is not necessary and duplicated. Fig 11 and 12 possibly should be in supplement. Upper part of fig 10 is hard to analyse and not necessary. Table 6 is quite large - consider please moving it to supplement. Similiary chapter 2.6 is not very conclusive and may be moved to supplement.
Generally paper is very interesting and uses advanced analysis of RNAseq
data, but is rather too long and some data should be better moved to
supplmnt to improve clarity of presentation. Descriprion of paragraph 2.9
need thorough revision. Why different reference genes were used for two
parts of qPCR analysis? Line 650 - what "All 8 transcripts"? line 655 - what
two genes?, lines 659-660 - genes in brackeets should not be pooled - data
should be analyzed statistically and description corrected. Fig 34 and 35
have the same captions - more details should be provided. Line 1075 check
table names, corret names of TIP1 in tables 9 and 8, correct abbreviation
line 1100, split line 1108. Fragment of discussion 3.5 should be integrated
with results. Finally, synthetic Fig. 42 needs better description in section
of disucission.
Author Response
Dear Reviewer of International Journal of Molecular Sciences:
Thank you very much for your review advises.
Responses to the review comments are as follows:
1. Introduction has been rewritten in a more ordered way.
2. About’ Explanation of abbreviations i.e. gene names……’
To verify the genes expression in our Illumina sequencing analyses, twenty DEGs containing two candidate genes (Gibberellic acid insensitive, GAI contig 1509; GA-insensitive dwarf mutant1a, GID1a contig_18004) related to the gibberellin signal transduction were selected for qRT-PCR using samples of 0h, 2h and 4h stages originally used for RNA-Seq, all of which are known to be related to Gibberellins signal, including the genes encoding GA2ox (gibberellin 2-oxidase, Contig_79537), DELLA (Dwarf8, Contig3159), CBF (C-repeat binding factor , Contig9007), LTCOR12 (gibberellic acid-stimulated in Arabidopsis, GASA, Contig_133755), ER (estrogen receptor, Contig_43645), AP2-EREBP (Apetala2/ethylene responsive element binding protein, Contig_6641), SAUR-like (small auxin-up RNA-like, Contig_96020), auxin-responsive GH3 (auxin-responsive Gretchen Hagen 3, Contig_120322), DELLA (GAI, Contig1509), GID1a (Contig18004) and so on. The Ct values of the LoTIP (tonoplast intrinsic protein) in all sample……
3. List of abbreviations have been extended.
FW Fresh weight;
ZR Zeatin riboside;
JA Jasmonic acid;
ABA Abscisic acid;
IAA Indole-3-acetic acid;
GA3 Gibberellic acid 3;
ELISA Enzyme-linked immuno sorbent assay;
Contig### First_Contig###;
contig_### 1-LWQ_no_rRNA.1_(paired)_contig_###
4. Caption of Fig 1 has been updated to a more clear one.
5. Fig 3-the original part A+B+C has been deleted.
6. Evaluation of RNA quality should be supplemented in methods section 4.3.
1. Tab 5, Tab 6, Fig 11 and Fig12 have been moved to supplement.
2. New captions have been added to Fig 34 and Fig 35.
Figure 20. qRT-PCR of 20 key genes selected from transcriptome data. (original Fig.34)
Figure 21. qRT-PCR of 10 key genes in GCN modules. (original Fig.35)
3. All names of contigs have been shortened and reduced to necessary unique abreviation (also in all figs, including Fig14-19, 24-27) have all been replaced by new ones), e.g. contig_53117 is an abbreviation for for 1-LWQ_no_rRNA.1_(paired)_contig_53117, Contig16608 is an abbreviation for First_Contig16608.
1. Figures in chapter 2.6 are all be moved to supplement.
2. Sequences of contigs mentioned in the MS will be deposited in some public database, and we need a few more days to finish it.
3. ‘All 8 transcripts’ (original line 650) changes to ‘All transcripts’.
4. Another gene of two genes (original line 655) is Contig1509 which is added to the above.
The results presented in Figure 34 showed that the expression levels of DELLA (Dwarf8 Contig3159, GAI Contig1509) decreased sharply in 2h and 4h samples than in 0h sample, which indicated that the two genes may play key roles in GA signal response and may delay the floral transition.
5. Genes in brackeets (original line 659-660) have been analyzed separately.
AP2-EREBP (contig_6641), SAUR-like (contig_96020) and auxin-responsive GH3 (contig_120322) are related to endogenous hormone. The expression of AP2-EREBP (contig_6641) continued to drop sharply in 2h, 4h and 8h, and increased a little in 16h, which was still less than half of the expression in 0h. The expression of SAUR-like (contig_96020) up-regulated dramatically in 2h, and continued to down-regulate in 4h and 8h, following a sharp recovery in 16h, which was almost 11 times the original expression level in 0h. The auxin-responsive GH3 (contig_120322) continued to up-regulate dramatically except a sharp fall in 8h.
6. Response to the question ‘why different reference genes were used for two parts of qPCR analysis?’ The qRT-PCR experiments verified that these kinds of references are all applicable in lilies, and that there is a duplicated gene AGL20 (contig_14431) demonstrating the same expression trend and level under the control of these different types of references.
7. About ‘synthetic Fig. 42 needs better description in section of disucission’.
According to transcriptome analysis and gene co-expression network analysis, the molecular regulation mechanism of gibberellin-induced flower induction in LA-lily 'Aladdin' was obtained (Figure 28).G-protein coupled receptor responding to the signal of exogenous GA3 activate several specific cation transport related genes encoding ammonium ion transporter AMT, plasma membrane ATPase, and calcium transport ATPase. The biosynthesis of endogenous gibberellin was activated, so extracellular gibberellin signals can be transferred to cells. From the very beginning, GGPP forms active GAs under the catalysis of a series of enzymes. Genes encoding KO, GA20ox, GA2ox, GA3ox, P450 etc. were found playing important roles in the process. Ca2+/CAM maybe the second messenger of gibberellin signal. Active GAs bind to the GID1 receptor, which allows subsequent interaction between GID1 and DELLA proteins to form the stable GID1-GA-DELLA complexes. Then the complexes are recognized by the SLY1 based SCFSLY1 complex, which ubiquitylates the DELLA proteins and causes their degradation by the 26S proteasome. The biological process is actually phosphorylation and glycosylation of DELLA protein. Many transcription factors such as WRKY7, WRKY21, WRKY22, NAC1, NAC2, NAC11, AP2, CBF, EREB1, CYP81 etc. specifically regulate this biological process. The DELLA protein’s phosphorylation and dephosphorylation process is reversible depending on the regulation of different transcription factors as shown in the figure below. The polysaccharide metabolism exists for supplying energy for the process. The reduction of DELLA promotes the transcription of downstream factor GASA which promotes the expression of SOC1(AGL20). Subsequently, the LFY expressed a lot to stimulate flowering. A research indicated that GA-induced expression of FT is dependent on CO to modulate flowering under LDs [19]. In our study, CO also expressed differentially during the exogenous GA3 treatment. Furthermore, epigenetic and stress-resisitent related genes and genes in the phytohormone metabolic pathway and L-phenylalanine metabolic pathway also participated the biological process.
1. The rest have been modified and supplemented as required.
I hope that I have addressed all the concerns in the previous review of the manuscript and that the integration of the reviewer's comments into the revised manuscript has substantially improved the manuscript.
Best regards to you
Sincerely yours
Yingmin Lyu

Reviewer 2 Report
In my opinion, the submitted paper by Li et al. is suitable for publication after changes specified below.
1. The total length of the manuscript is very large. Part of the data is better transferred to the Supplementary files: Fig. 11, 12 (just mentioned in the text), Fig. 14-19, 23, 25, 27, 29, 31, 33 (in the text is listed the contents of the figures).
2. In the Introduction the authors mention studies on the transcription factors of GA pathway in lily (L.124-125). It is advisable to provide a brief description and references.
3. Why did not analyze the transcriptome after 8 and 16 h GA treatment?
4. Various dimensions in the text: mg l-1 (L.69), mg L-1 (L.70), mg/L (L.16, 36, etc.).
5. Plant names should be in italics (L. 57, 68, etc.)
6. The Conclusions is present only in the Abstract, but not in the paper.
7. References: a) no volume and page numbers (Ref. 1, 5, 11, etc.); b) only one author of the two is present (Ref. 5); c) description of references does not meet the Instructions for authors.
More specific comments:
L.154-163. The text lists the data from Fig. 2 and should be reduced.
Fig. 4. a) % data differ from the data in Table 4 and in the text (e.g. between 12.9 and 14.51% for “T” vs. 17.36% in Table and L.213, etc.; b) citation of Fig. 4 is absent in the text.
L.203. Table 2 is an error probably should be Table 3.
L.230. Fig. 2 is an error probably should be Fig. 5.
Fig. 5: a) the pie charts should be named; b) the pie charts and the font of the captions must be the same size; c) pie charts 1 and 2 correspond to the BP and SS, but chart 3 is part of the BP and SS, but not MF. For this reason there are errors in the text (L.221-223): the “cell” refers to CC, but not MF, the “single-organism process” refers to BP, but not MF.
L.243. Fig. 3 is an error probably should be Fig. 6.
L.272. Fig. 7 should be cited after “respectively”.
Fig. 7. “3 vs 1” is better to replace “4 h vs 0 h”, etc.
Fig. 10: a) % data for BP should be moved beyond the border of the pie chart (as in CC and MF); b) some of the signatures to MF pie chart are missing, c) swap CC and MF pie charts, as in the text and in the earlier figures the order is as follows: BP - CC - MF.
L.454. Table number is absent.
Fig. 39-41. There are inscriptions in Chinese.
L. 556. “Fig 3C and Table 7 Suppl” - ? Supplement in the manuscript is missing.
L. 1075: a) Tab should be replaced by Table; b) Table 1 and 2 are an error probably should be Table 8 and 9.
Author Response
Please open the PDF document

Reviewer 3 Report
This is an engaging article with robust methodology However, the presentation of results is somewhat confusing, and the readability of the discussion could be improved. Addressing both these issues will make this paper more impactful.
Abstract
& : The aim is very clear, the tilte is informative
Title
Introduction: The research question clearly outlined avery good introduction
Methods : Study methods valid and reliable
Results: Data presented in an appropriate way except for some figures
Discussion : Results discussed from multiple angles and placed into context without being overinterpreted
Major Point: I think that it should more explained these data instead of( were found)they were up regulated or Down regulated?
. 29 In addition, stress resistance related genes such as LEA1, LEA2, LEA4, serine/threonine protein
. 30 kinase, LRR receptor‐like serine/threonine protein kinase, P34 kinase, histidine Kinase 3 and epigenetic related
. 31 genes in DNA methylation, histone methylation, acetylation, ubiquitination of ribose were also found .
. 32 Particularly, a large number of transcription factors responsive to the exogenous gibberellin signal including
. 33 WRKY40, WRKY33, WRKY27, WRKY21, WRKY7, MYB, AP2/EREBP, bHLH, NAC1, NAC2, NAC11 were found
I am alittle bit confused to understand the following Data and i think there is a contradiction between DEGs 2h and 0h in first data and the 2h and 0h in the second one:
. 243 This suggests that the DEGs between 4h and 0h
. 244 is larger than that between the 2h and 0h libraries, while the difference between the 4h and 2h libraries is the
. 245 smallest of the three.
.
. 251 for 8% of the total DEGs. The total DEGs between the 2h and 0h libraries accounted for 54.6% of the total DEGs,
. 252 suggesting that genes expressed rapidly in response to GA3 treatment.
.
Minor points :
· Insert the unit of the content for figure 1
· Identify each circle in the chart as it written in the table figure 5
· poor quality of the photos for Figures 8,10,21
· I twill better to transfer the description of the following figures to
be under the figure instead of up figure 22 to 33
Author Response
Please open the PDF document
